# Neighborhood Reconstructing Autoencoders

**Yonghyeon Lee**[1]     **Hyeokjun Kwon**[1]     **Frank C. Park**[1,2]
Seoul National University[1]     Saige Research[2]
{yhlee, hj.kwon}@robotics.snu.ac.kr     fcp@snu.ac.kr

## Abstract

Vanilla autoencoders often produce manifolds that overfit to noisy training data, or have the wrong local connectivity and geometry. Autoencoder regularization techniques, e.g., the denoising autoencoder, have had some success in reducing overfitting, whereas recent graph-based methods that exploit local connectivity information provided by neighborhood graphs have had some success in mitigating local connectivity errors. Neither of these two approaches satisfactorily reduce both overfitting and connectivity errors; moreover, graph-based methods typically involve considerable preprocessing and tuning. To simultaneously address the two issues of overfitting and local connectivity, we propose a new graph-based autoencoder, the **Neighborhood Reconstructing Autoencoder (NRAE)**. Unlike existing graph-based methods that attempt to encode the training data to some prescribed latent space distribution – one consequence being that only the encoder is the object of the regularization – NRAE merges local connectivity information contained in the neighborhood graphs with local quadratic approximations of the decoder function to formulate a new **neighborhood reconstruction loss**. Compared to existing graph-based methods, our new loss function is simple and easy to implement, and the resulting algorithm is scalable and computationally efficient; the only required preprocessing step is the construction of the neighborhood graph. Extensive experiments with standard datasets demonstrate that, compared to existing methods, NRAE improves both overfitting and local connectivity in the learned manifold, in some cases by significant margins. Code for NRAE is available at https://github.com/Gabe-YHLee/NRAE-public.

## 1 Introduction

Autoencoders are widely used to identify, and to generate samples from, the underlying low-dimensional manifold structure of a given data distribution [14, 1]. It has been widely observed that vanilla autoencoders quite often produce manifolds that (i) are highly sensitive to noisy training data (see Figure 1(a)), or (ii) have the wrong local connectivity and geometry (see Figure 1(b)), significantly impairing their performance. Regularization techniques have had some success in mitigating the former, e.g., the Denoising Autoencoder [21], which uses deliberately corrupted inputs to train the autoencoder, typically learns manifolds that are robust to noise, but not always with the correct local geometry.

Recently, autoencoder regularization methods that use neighborhood graphs have had some success in addressing the incorrect connectivity issue [16, 18, 10, 7]. Notwithstanding the additional computational overhead of constructing local neighborhood graphs, the local geometric information obtained from these graphs can significantly reduce errors in the local geometry of the learned manifold, and make learning more well-behaved and robust.

The underlying premise behind these methods is that since the local geometry and topology of the data is captured in the latent space distribution, which is determined entirely by the encoder, regularizing only the encoder should be sufficient; little if any consideration needs to be given to the decoder.

35th Conference on Neural Information Processing Systems (NeurIPS 2021).

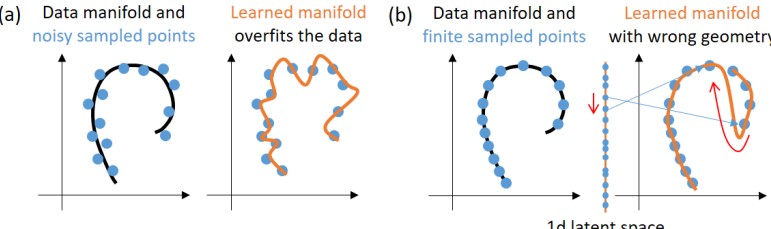

Figure 1: Learned manifolds that (a) overfit the data or (b) have the wrong local geometry.

The flaw with this premise is that although the encoder learns the correct latent space representation of the manifold data, the decoder is still susceptible to overfitting of the type shown in Figure 1(a). These existing methods moreover rely on computation-intensive preprocessing steps like manifold learning [10], linear coefficients computation [7], or computing topological features using persistent homology at each training iteration [16, 2], each of whose computational requirements can grow significantly with problem dimension and scale.

The main contribution of this paper is a new graph-based autoencoder training method that addresses both the overfitting and connectivity issues illustrated in Figure 1(a)-(b). Like current methods, our method also employs local graphs that capture the local geometry of the data distribution. The key idea behind our method, which we call the **Neighborhood Reconstructing Autoencoder (NRAE)**, is to employ a local quadratic (and in some cases linear) approximation of the decoder function to formulate a new **neighborhood reconstruction loss** in lieu of the point reconstruction loss typically used for autoencoder training. This idea leads to learning the correct geometry and reducing noise sensitivity, significantly improving the robustness of autoencoder training.

To make things more explicit, let $g_\phi : \mathbb{R}^n \to \mathbb{R}^m$ be the encoder (parametrized by $\phi$) and $f_\theta : \mathbb{R}^m \to \mathbb{R}^n$ be the decoder (parametrized by $\theta$). Whereas vanilla autoencoders are trained to minimize the sum of the point reconstruction errors $\sum_i \|x_i - f_\theta(g_\phi(x_i))\|^2$, NRAE minimizes a reconstruction error of the form

$$\sum_i \sum_{x \in \mathcal{N}(x_i)} \left\| x - \tilde{f}_\theta(g_\phi(x); g_\phi(x_i)) \right\|^2, \tag{1}$$

where $\mathcal{N}(x_i)$ is the set of neighborhood points of $x_i$ (including $x_i$) and $\tilde{f}_\theta(\cdot; g_\phi(x_i))$ is a local quadratic (or linear) approximation of $f_\theta$ about $g_\phi(x_i)$. The vanilla autoencoder is obtained by setting $\mathcal{N}(x_i) = \{x_i\}$ for all $i$. The key idea here is to locally approximate the decoder only, and to exploit the local geometric information extracted from the decoded manifold represented by the image of $f_\theta$.

Like other neighborhood graph-based methods, NRAE also learns the correct local geometry of the decoded manifold. At the same time, the local quadratic (or linear) approximation of $f_\theta$ considerably reduces any overfitting to noisy training data or sensitivity to outliers, while maintaining computational efficiency – rather than the entire Jacobian or Hessian of $f_\theta$, only the more easily computed Jacobian-vector and Hessian-vector products are needed for the approximation.

Compared to existing graph-based autoencoder regularization methods, NRAE is easy to implement, computationally efficient, and scalable, requiring only a single prior construction of the graph without additional pre-processing steps. Experiments with both synthetic and image data (*MNIST, Fashion-MNIST, KMNIST, Omniglot, SVHN, CIFAR10, CIFAR100, CelebA*) confirm that overall our method better learns the correct geometry of manifolds, showing improved generalization performance vis-á-vis existing graph-based and other autoencoder regularization methods.

## 2 Neighborhood Reconstructing Autoencoder

In this section, we first provide a high-level mathematical description of the Neighborhood Reconstructing Autoencoder (NRAE), followed by algorithmic details and a discussion of the NRAE's properties and behavior. Throughout we consider a deterministic autoencoder with an encoder function $g_\phi : \mathbb{R}^n \to \mathbb{R}^m$ and decoder function $f_\theta : \mathbb{R}^m \to \mathbb{R}^n$ ($m \leq n$), with their composition denoted by $F_{\theta,\phi} := f_\theta \circ g_\phi$. We use the notation $\mathcal{D} := \{x_i \in \mathbb{R}^n\}_{i=1}^M$ to denote the set of observed data points.

## 2.1 Mathematical Description

In what follows we use the notation $\mathcal{N}(x)$ to denote the set of neighborhood points of $x$, with $x$ included in $\mathcal{N}(x)$. We begin with the following definition:

**Definition 1** *Let $\tilde{F}_{\theta,\phi}(\cdot;x) := \tilde{f}_\theta(g_\phi(\cdot);g_\phi(x))$, where $\tilde{f}_\theta(\cdot;z)$ is a local quadratic (or in some cases linear) approximation of $f_\theta$ at $z = (z^1, z^2, ...z^m)$:*

$$\tilde{f}_\theta(z';z) := f_\theta(z) + \sum_{i=1}^{m} \frac{\partial f_\theta}{\partial z^i}(z)dz^i + \sum_{i,j=1}^{m} \frac{1}{2}\frac{\partial^2 f_\theta}{\partial z^i \partial z^j}(z)dz^i dz^j, \tag{2}$$

*where $dz = z' - z$. $\tilde{F}_{\theta,\phi}(\mathcal{N}(x);x)$ is said to be a **neighborhood reconstruction** of $\mathcal{N}(x)$.*

If instead of a quadratic approximation we use the linear approximation of $f_\theta$, the image of $\tilde{F}_{\theta,\phi}(\cdot;x)$ is the tangent space of the decoded manifold at $F_{\theta,\phi}(x)$, and the neighborhood reconstruction of $\mathcal{N}(x)$ is a subset of the tangent space; the neighborhood reconstruction in this case contains first-order local geometric information about the decoded manifold.

The key idea behind Definition 1 is that we locally approximate the decoder, and not the encoder, to extract and exploit local geometric information on the decoded manifold, which is captured in the image of $\tilde{F}_{\theta,\phi}(\cdot;x)$ (i.e., the local approximation of the decoded manifold). Figure 2 illustrates an example where the autoencoder reconstructs the points almost perfectly, but the neighborhood reconstruction of $\mathcal{N}(x)$, whose elements lie in the tangent space (here we use the linear approximation of $f_\theta$) is considerably different from $\mathcal{N}(x)$.

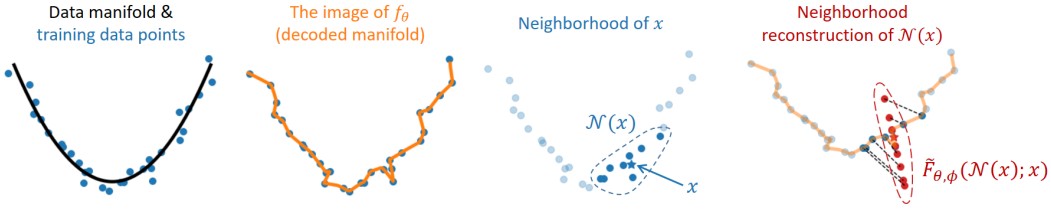

Figure 2: The training data points (blue), the decoded manifold (orange), the neighborhood of $x$ denoted by $\mathcal{N}(x)$, and the neighborhood reconstruction (red). The black dotted lines represent the correspondences between $x' \in \mathcal{N}(x)$ and $\tilde{F}_{\theta,\phi}(x';x)$.

Given that the neighborhood reconstruction of $\mathcal{N}(x)$ reflects the local geometry of the decoded manifold, minimizing a loss function that measures the difference between $\mathcal{N}(x)$ and its image $\tilde{F}_{\theta,\phi}(\mathcal{N}(x);x)$ is one means of training an autoencoder to preserve the local geometry of the original data distribution. With that goal in mind, we formulate a **neighborhood reconstruction loss** $\mathcal{L}$ as follows:

$$\mathcal{L}(\theta,\phi;\mathcal{D}) = \frac{1}{|\mathcal{D}|}\sum_{x\in\mathcal{D}}\frac{1}{|\mathcal{N}(x)|}\sum_{x'\in\mathcal{N}(x)} K(x',x) \cdot \|x' - \tilde{F}_{\theta,\phi}(x';x)\|^2, \tag{3}$$

where $K(x',x)$ is a positive symmetric kernel function that determines the weight for each $x' \in \mathcal{N}(x)$. Figure 3 illustrates how the neighborhood reconstruction loss can differentiate among the quality of the learned manifolds whose point reconstruction losses are all the same (close to zero): Case 3 has the smallest neighborhood reconstruction loss compared to Case 1 (wrong local geometry) and Case 2 (overfitting). NRAE converges to the vanilla AE – that is, the neighborhood reconstruction loss recovers the point reconstruction loss – if one of the following conditions is met: (i) $\mathcal{N}(x) = \{x\}$, (ii) $K(x',x) = \delta(x',x)$, or (iii) $f_\theta$ is linear.

It is reasonable to ask whether there are any advantages to using an approximation for both the encoder and decoder, i.e., to use a local quadratic (or linear) approximation for the composition map $F_{\theta,\phi}$ rather than just the decoder. As verified below in Section 4.3, using a quadratic approximation for both the encoder and decoder results in minimal to no performance improvement (the results for this Extended NRAE (E-NRAE) case are nearly identical to those obtained for NRAE), but the computational requirements increase substantially. As intuition suggests, applying a quadratic approximation for the decoder is sufficient in evaluating the neighborhood reconstruction loss.

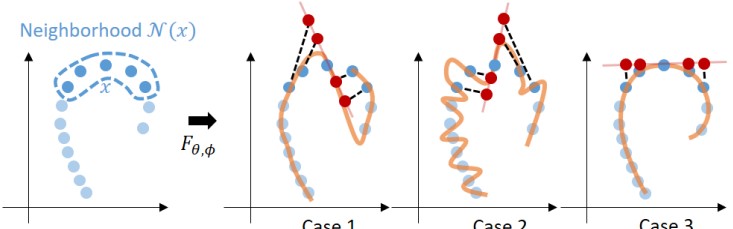

Figure 3: The orange curves represent the learned manifolds, the red points represent the neighborhood reconstruction, and the lengths of the black dotted lines represent the neighborhood reconstruction loss.

## 2.2 Algorithmic Details

**Graph construction.** The problem of inferring the geometric structure of a data distribution is typically posed as a graph construction problem [4]. We use one of the simplest graph construction methods, the k-NN graph with the Euclidean distance metric. The robustness of our algorithm with respect to the choice of $k$ is tested in the Supplementary Material.

**Kernel design.** We choose the following simple kernel

$$K(x', x) = \lambda + (1 - \lambda)\, \delta(x', x), \tag{4}$$

where $0 \leq \lambda < 1$ and $\delta(x', x) = 1$ if $x' = x$ and zero otherwise. This assigns the weight 1 for the center $x \in \mathcal{N}(x)$ and the weight $\lambda$ for the remaining neighborhood points.

**Batch sampling.** To estimate the gradient of the proposed loss function, we use batch sampling for both summations over $\mathcal{D}$ and $\mathcal{N}(x)$. Given a batch $\mathcal{B} \subset \mathcal{D}$, we again sample a batch $\mathcal{B}_x$ from $\mathcal{N}(x)$. We empirically find that forcing each batch $\mathcal{B}_x$ to include $x$ improves convergence. In this paper, we set $\mathcal{B}_x = \{x, x_n\}$ where $x_n$ is uniformly sampled from $\mathcal{N}(x) - \{x\}$.

## 3 Related Work: Regularization of Autoencoders

In this section, we review some standard autoencoder regularization techniques and their relation to NRAE: i) regularizing latent space distributions, ii) Jacobian regularization, and iii) regularization using neighborhood graphs.

**Regularizing latent space distributions.** One popular regularization strategy is to enforce the latent space distribution (an approximate posterior) to be close to some user-specified prior distribution; some examples include the Variational Autoencoder (VAE) [12], Adversarial Autoencoder (AAE) [15], and Wasserstein Autoencoder (WAE) [19]. The Gaussian distribution is a popular choice for the prior, but this choice often leads to over-regularization. Several works improve the VAE by learning more complex priors, or an optimal prior in terms of maximizing the training objective function of the VAE [8, 6, 20, 13]. These techniques are developed to make latent space distributions close to some easy-to-sample prior distribution, but are not designed to resolve the two main issues (local geometry and overfitting) addressed in this paper.

**Jacobian regularization.** The Contractive Autoencoder (CAE) attempts to enhance robustness of representation by penalizing the Jacobian norm of the encoder function [17]. However, like other encoder-only regularization methods, it often learns a manifold that overfits the data. More recently [5] regularizes the Jacobian of the decoder function to learn a flat manifold. The trained decoder function may produce a smooth manifold but with the wrong local geometry. The main difference between these methods and NRAE is the use of a neighborhood graph containing local connectivity information.

**Regularization using neighborhood graphs.** Several recent works have developed graph-based autoencoder regularization methods that, at least implicitly, try to learn manifolds with the right local geometry: Generalized Autoencoder (GAE) [22], Topological Autoencoder (TopoAE) [16],

Witness Autoencoder (W-AE) [18], Geometry Regularized Autoencoder (GRAE) [10], and Structure-Preserving Variational Autoencoder (Sp-VAE) [7]. These works focus almost entirely on the encoder when regularizing the latent space distribution (with the exception of GAE [22]); using the regularization term that does not depend on the decoder lead to noise sensitivity and overfitting issues as described in detail above. These methods are also computationally intensive, especially for large scale high-dimensional problems, as a result of expensive pre-processing [10, 7] or computation in training [16, 18].

In contrast, by focusing more on the geometry of the decoded manifold, NRAE is able to learn smooth manifolds with the correct local geometry, simpler to implement, and more scalable since it requires only a single prior construction of the graph without other additional pre-processing steps. Indeed, with the exception of Sp-VAE [7], all other approaches test their algorithms with two-dimensional latent spaces, while we test NRAE for latent spaces with up to 128 dimensions.

For general regularization methods that focus on latent space distributions, we surmise that using our neighborhood reconstruction loss in lieu of the point reconstruction loss may lead to several performance improvements; this is verified in Section 4.3, where we combine our neighborhood reconstruction loss with some existing encoder regularization methods and compare their performance against the original methods.

# 4 Experiments

In this section, through extensive experiments with both synthetic and real-world image data, we compare NRAE with a range of existing regularization methods: the Variational Autoencoder (VAE) [12], Wasserstein Autoencoder (WAE) [19], Denoising Autoencoder (DAE) [21], Contractive Autoencoder (CAE) [17], Geometry Regularized Autoencoder (GRAE) [10], and Structure-Preserving Variational Autoencoder (Sp-VAE) [7]. For comparison with Sp-VAE, we augment the regularization term introduced in [7] to a vanilla autoencoder rather than the variational autoencoder – we refer to this autoencoder as the Structure-Preserving Autoencoder (SPAE) – since a direct comparison with the variational autoencoder is already made, and our intent is to examine the effects of the structure-preserving regularization term.

For NRAE, we use both the local quadratic and linear approximations, respectively denoted NRAE-Q and NRAE-L. Our focus is on comparing the manifold smoothing property, geometry preserving property, and generalization ability of NRAE against baseline methods. We refer the reader to the Supplementary Material for a description of the network architectures used in the experiments, together with implementation details including the hyperparameter tuning strategy.

## 4.1 Manifold Smoothing Property

**Smoothness of learned manifolds.** We first consider a one-dimensional manifold embedded in $\mathbb{R}^2$, $\{x, \sin(x)|x \in [-\pi, \pi]\}$. Given 50 randomly sampled data points, we add isotropic Gaussian noise with a standard deviation of 0.2. We train the NRAE and baseline AEs with one-dimensional latent spaces. Figure 4 and Table 1 show that NRAE-L and NRAE-Q successfully re-produce smooth manifolds compared to other baseline methods.

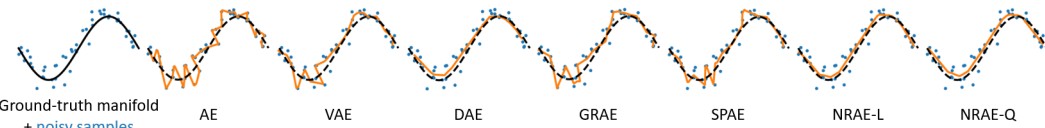

Figure 4: The noisy samples (blue) and learned manifolds (orange).

Table 1: The mean-squared reconstruction errors of 10,000 clean test data. The averages and standard deviations are computed over ten times run with different noises, and multiplied by 100 (the highest and lowest scores are ignored). The best and second-best results are colored red and blue, respectively.

| AE | VAE | WAE | DAE | CAE | GRAE | SPAE | NRAE-L | NRAE-Q |
|---|---|---|---|---|---|---|---|---|
| $1.90 \pm 0.23$ | $1.45 \pm 0.39$ | $2.02 \pm 0.76$ | $1.09 \pm 0.21$ | $1.33 \pm 0.30$ | $1.56 \pm 0.43$ | $1.28 \pm 0.30$ | $0.29 \pm 0.06$ | $0.30 \pm 0.07$ |

Second, we rotate 100 MNIST images of the digit eight 100 times by 3.6 degrees, obtaining a set of 10,000 training data. We train the NRAE and baseline AEs with two-dimensional latent spaces, normalize each latent space, and generate rotating images by sampling regular grids of the latent spaces. NRAE-L and NRAE-Q generate smoothly varying images compared to other baselines (Figure 5). The DAE does not generate smoothly varying samples because the noise statistics (Gaussian noise) used in training is different from the statistical noise in the rotating image data.

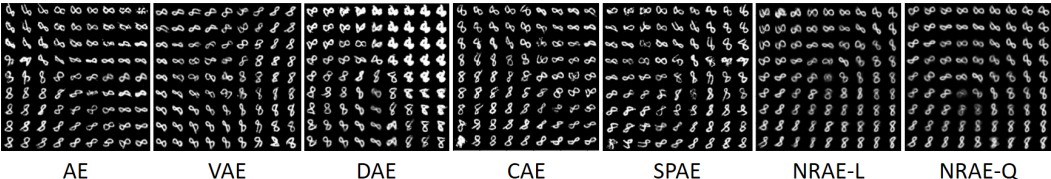

Figure 5: Generated samples of rotating digit 8 from regular grids of the latent spaces.

Finally, to confirm the denoising effect of NRAE on common image data, we train the NRAE and baseline AEs on MNIST and CIFAR10 data corrupted with various levels of Gaussian noise (we use sufficiently large epochs for convergence). We then compare denoising performance measured by the Peak Signal-to-Noise Ratio (PSNR). As shown in Table 2, NRAE-L and NRAE-Q outperform other baselines including DAE which performed the third best (the noise statistics used in training is the same as the added noise statistics).

Table 2: Comparison of PSNR (higher-the-better). The latent space dimensions are 16 for MNIST and 128 for CIFAR10. The best and second-best results are colored red and blue, respectively.

| Dataset | Noise | AE | VAE | WAE | DAE | CAE | GRAE | SPAE | NRAE-L | NRAE-Q |
|---------|-------|-------|-------|-------|-------|-------|-------|-------|--------|--------|
| MNIST | 0.1 | 20.19 | 19.61 | 19.83 | 20.69 | 20.08 | 20.06 | 20.01 | 21.07 | 21.27 |
|       | 0.2 | 17.15 | 16.36 | 16.96 | 18.37 | 17.24 | 17.34 | 17.34 | 18.55 | 18.90 |
| CIFAR10 | 0.1 | 18.22 | 19.87 | 19.54 | 20.42 | 20.73 | 19.79 | 19.43 | 22.21 | 22.27 |
|        | 0.2 | 17.84 | 18.25 | 17.38 | 19.30 | 18.86 | 16.98 | 16.90 | 21.04 | 20.94 |

**Curvature of learned manifolds.** We train the NRAE and baseline AEs on MNIST images with two-dimensional latent spaces, normalize each latent space, then visualize the scalar curvature field (i.e., twice the Gaussian curvature [9]) (Figure 6). As shown in Figure 6, NRAE-L and NRAE-Q both learn flatter manifolds compared to our baseline AEs.

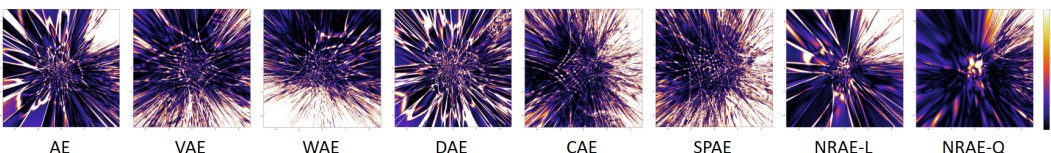

Figure 6: The scalar curvature field (brighter-the-larger).

## 4.2 Geometry Preserving Property.

**Swiss roll.** Consider a one-dimensional Swiss roll $r(\theta) = 0.1 + 0.9 \cdot \theta/(2\pi)$ for $\theta \in [0, 2\pi]$. Given 30 randomly sampled data points, we add isotropic Gaussian noise with a standard deviation of 0.01. We then train the NRAE and baseline AEs with one-dimensional latent spaces. Figure 7 shows the reconstruction results for 1000 test data points. Only GRAE, SPAE, and NRAE-Q successfully learn manifolds with the right local geometry. However, unlike NRAE-Q, the regularization terms in GRAE and SPAE have the effect of increasing reconstruction errors (observe the red rectangles in Figure 7). NRAE-L fails to reconstruct points around the point $x_r$ circled in red: the nearest neighbor to $x_r$ in the training set, marked by the blue circle, lies close to the tangent space of the decoded manifold at $x_r$, i.e., the neighborhood reconstruction loss is small.

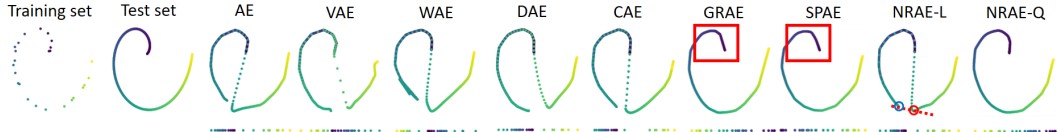

Figure 7: Swiss roll data trained with one-dimensional latent spaces. The dots below the figures represent the one-dimensional latent space encoding of the training set.

**Rotated/Shifted MNIST.** First, as shown in the first row of the two figures in Figure 8, we generate two sets of 20 training data: the rotated MNIST images of the digit 3 and shifted MNIST images of the digit 7. We train NRAE and baseline AEs with one-dimensional latent spaces, encode the training data to the latent spaces, sort the encoded values in ascending order, and decode them to generate the images (Figure 8). Only SPAE, NRAE-L, and NRAE-Q successfully learn manifolds of the rotated/shifted images with the correct geometry.

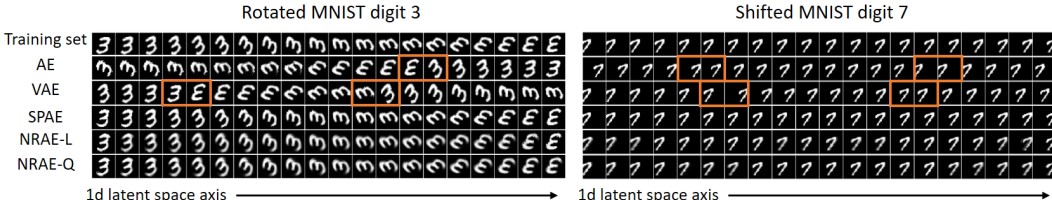

Figure 8: Generated rotated/shifted MNIST images. Discontinuities are marked by orange boxes.

Second, we rotate an MNIST image of the digit six 300 times by 1.2 degrees and obtain 300 training data; 40 of these are visualized in the first row of Figure 9. We further generate 1000 test data in a similar manner. We then train the AE, DAE, CAE, and NRAE with one-dimensional circular latent spaces by adding one layer to the end of the encoder corresponding to $z \rightarrow z/\|z\|$, and visualize which data are encoded to which point in the latent spaces via the cyclic color map. Only NRAE-Q is able to learn a manifold with the correct local geometry.

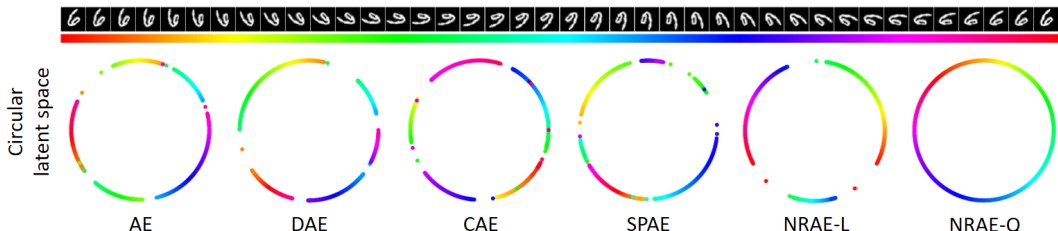

Figure 9: Circular latent space encoding of the rotated MNIST images of digit 6.

### 4.3 Generalization

**Test data reconstruction.** With various large-scale standard benchmark image data, we show that NRAE generalizes better to the test data compared to other baseline methods. We use the test reconstruction Mean Square Error (MSE) for quantitative comparison.

First, we train fully-connected neural networks with the MNIST, FMNIST, and KMNIST data by varying the number of training data from $1000, 2000 \ldots, 10,000$, and compare the test reconstruction MSEs. The numbers for the validation and test data are fixed at $10,000$ and $50,000$, respectively. Figure 10 shows the MSEs as a function of the number of training data. The MSEs decrease as the number of training data increases, while those of the NRAE are lower than most of the other baselines regardless of the number of training data.

Second, we train convolutional neural networks and compare the test reconstruction MSEs (Table 3). We conduct two experiments i) with the entire public data denoted by L (large) and ii) with the subset

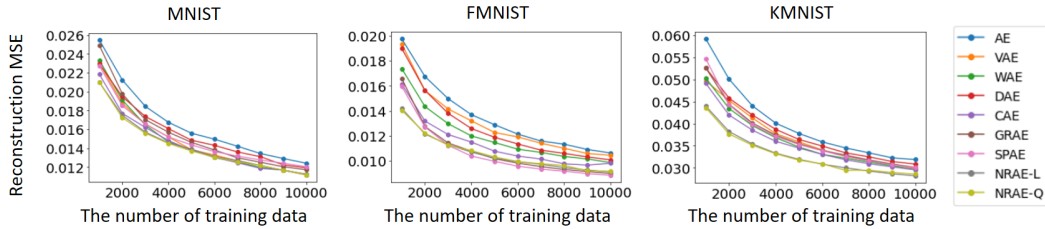

Figure 10: The test reconstruction MSEs as the number of training data changes.

of data denoted by S (small). The number of selected subsets are around $20 \sim 30\%$ of the entire data. NRAE-L and NRAE-Q produce lower reconstruction MSEs than most of the other baselines, meaning that the manifolds learned with our methods better generalize to the test data. The CAE experiment on CELEBA data is excluded because of its high computational cost for high-dimensional data. Other measures such as the Frechet-Inception Distance (FID) scores [11] and the Evidence Lower Bound (ELBO) are reported in the Supplementary Material.

Table 3: The test reconstruction MSEs, the lower the better. The latent space dimensions are 16, 32, 32, 32, 64, 128, 128, 128 for MNIST, FMNIST, KMNIST, Omniglot, SVHN, CIFAR10, CIFAR100, and CELEBA, respectively. The best and second-best results are colored red and blue, respectively.

| Dataset | Size | AE | VAE | WAE | DAE | CAE | GRAE | SPAE | NRAE-L | NRAE-Q |
|---------|------|---------|---------|---------|---------|---------|---------|---------|---------|---------|
| MNIST | S | 0.01002 | 0.01091 | 0.01009 | 0.00999 | 0.00998 | 0.01004 | 0.00989 | 0.00953 | 0.00968 |
|  | L | 0.00688 | 0.00756 | 0.00690 | 0.00684 | 0.00692 | 0.00696 | 0.00694 | 0.00649 | 0.00683 |
| FMNIST | S | 0.01485 | 0.01652 | 0.01428 | 0.01446 | 0.01319 | 0.01331 | 0.01363 | 0.01289 | 0.01277 |
|  | L | 0.01118 | 0.01235 | 0.01106 | 0.01099 | 0.01052 | 0.01060 | 0.01065 | 0.01060 | 0.01044 |
| KMNIST | S | 0.03267 | 0.03234 | 0.03283 | 0.03280 | 0.03279 | 0.03206 | 0.03268 | 0.03071 | 0.03021 |
|  | L | 0.02844 | 0.02963 | 0.02776 | 0.02814 | 0.02762 | 0.02753 | 0.02732 | 0.02564 | 0.02602 |
| Omniglot | S | 0.03038 | 0.03627 | 0.03078 | 0.03068 | 0.02714 | 0.02967 | 0.02889 | 0.02668 | 0.02631 |
|  | L | 0.02704 | 0.03192 | 0.02728 | 0.02696 | 0.02567 | 0.02648 | 0.02644 | 0.02578 | 0.02539 |
| SVHN | S | 0.00320 | 0.00420 | 0.00320 | 0.00369 | 0.00273 | 0.00317 | 0.00307 | 0.00202 | 0.00192 |
|  | L | 0.00174 | 0.00204 | 0.00190 | 0.00177 | 0.00178 | 0.00173 | 0.00175 | 0.00148 | 0.00147 |
| CIFAR10 | S | 0.01466 | 0.01620 | 0.01431 | 0.01427 | 0.01208 | 0.01452 | 0.01504 | 0.00768 | 0.00691 |
|  | L | 0.00960 | 0.01123 | 0.00863 | 0.00900 | 0.00755 | 0.00832 | 0.00898 | 0.00629 | 0.00587 |
| CIFAR100 | S | 0.01465 | 0.01713 | 0.01463 | 0.01484 | 0.01369 | 0.01391 | 0.01477 | 0.00765 | 0.00717 |
|  | L | 0.01015 | 0.01064 | 0.00951 | 0.00862 | 0.00842 | 0.00910 | 0.00912 | 0.00678 | 0.00635 |
| CELEBA | S | 0.00780 | 0.00937 | 0.00830 | 0.00782 | - | 0.00814 | 0.00861 | 0.00608 | 0.00747 |
|  | L | 0.00613 | 0.00646 | 0.00630 | 0.00590 | - | 0.00595 | 0.00665 | 0.00563 | 0.00565 |

**Compatibility with other regularization methods.** In this section, we show that the NRAE loss function can be used together with other existing latent space distribution regularization methods to improve their generalization performance. We train convolutional neural networks with the MNIST and CIFAR10 data (small). As shown in Table 4, the NRAE loss improves WAE, CAE, and SPAE by significant margins. For VAE, there is only little improvement, primarily because the VAE regularization term is too dominant in training. Interestingly, some cases of WAE, CAE, and SPAE combined with the NRAE loss show even better generalization performance compared to NRAE alone. These results imply that a proper combination of the neighborhood reconstruction loss and existing regularization methods can lead to further improvements in performance.

**Extended NRAE.** Instead of using a quadratic (or linear) approximation of $f_\theta$, we can approximate the composition function $F_{\theta,\phi} = f_\theta \circ g_\phi$ to define the neighborhood reconstruction loss. While NRAE only requires the computation of the Jacobian-vector product or Hessian-vectors product of $f_\phi$, E-NRAE requires the computation of these quantities for $F_{\theta,\phi}$. We train convolutional neural networks on the MNIST and CIFAR10 data (small) and compare their generalization abilities and computational requirements. As shown in Table 5, the extended versions show comparable generalization performance to the original version, yet take a longer per-epoch runtime (100 batch size and 10,000 training data).

Table 4: Comparison of the test reconstruction MSEs for VAE, WAE, CAE, GRAE, and SPAE before and after being combined with the NRAE loss function. MNIST and CIFAR10 data (small) are used.

| Dataset | NRAE-L | NRAE-Q | VAE | NRVAE-L | NRVAE-Q | WAE | NRWAE-L | NRWAE-Q |
|---|---|---|---|---|---|---|---|---|
| MNIST | 0.00953 | 0.00968 | 0.01091 | 0.01089 | 0.01098 | 0.01009 | 0.00952 | 0.00966 |
| CIFAR10 | 0.00768 | 0.00691 | 0.01620 | 0.01613 | 0.01609 | 0.01431 | 0.00707 | 0.00684 |

| | NRAE-L | NRAE-Q | CAE | NRCAE-L | NRCAE-Q | SPAE | NRSPAE-L | NRSPAE-Q |
|---|---|---|---|---|---|---|---|---|
| MNIST | 0.00953 | 0.00968 | 0.00998 | 0.00936 | 0.00965 | 0.00989 | 0.00942 | 0.00975 |
| CIFAR10 | 0.00768 | 0.00691 | 0.01208 | 0.00718 | 0.00724 | 0.01504 | 0.00723 | 0.00703 |

| | NRAE-L | NRAE-Q | GRAE | NRGRAE-L | NRGRAE-Q | |
|---|---|---|---|---|---|---|
| MNIST | 0.00953 | 0.00968 | 0.01004 | 0.00945 | 0.00974 | |
| CIFAR10 | 0.00768 | 0.00691 | 0.01452 | 0.00716 | 0.00694 | |

Table 5: Comparison of the test reconstruction MSEs and per-epoch runtime estimates for NRAE-L, NRAE-Q, E-NRAE-L, and E-NRAE-Q using MNIST and CIFAR10 data (small).

| Dataset | Metric | NRAE-L | E-NRAE-L | NRAE-Q | E-NRAE-Q |
|---|---|---|---|---|---|
| MNIST | mse | 0.00953 | 0.01000 | 0.00968 | 0.01017 |
| | runtime | 21.57 s | 24.85 s | 34.73 s | 40.96 s |
| CIFAR10 | mse | 0.00768 | 0.00712 | 0.00691 | 0.00732 |
| | runtime | 59.14 s | 61.28 s | 89.24 s | 100.18 s |

## 5 Conclusion

This paper has proposed a new graph-based autoencoder, the Neighborhood Reconstructing Autoencoder (NRAE), that is capable of learning accurate manifolds that are robust to noisy training data and have the correct local connectivity and geometry, while being easy to implement, scalable, and computationally efficient. Neighborhood graphs that capture the local geometry of the data distribution are combined with local quadratic (or linear) approximations of the decoder function to formulate a new neighborhood reconstruction loss, which turns out to be a generalization of the original point reconstruction loss. Through extensive experiments with both synthetic and standard image datasets, we have demonstrated the manifold smoothing property, geometry preserving property, and the generalization performance advantages – in some cases by significant margins – of our method.

Further, we have empirically verified that (i) a proper combination of the neighborhood reconstruction loss and existing regularization terms that focus on the latent space distributions can lead to further improvements in performance and (ii) in neighborhood reconstruction loss, approximating the decoder only is sufficient for learning the correct manifold while being computationally more efficient than approximating both the encoder and decoder.

Our algorithm can be further enhanced in a number of different ways. First, the current implementation of NRAE uses the k-NN graph construction with Euclidean distance metric and a simple kernel function that outputs binary values. These choices are made for simplicity, and clearly sub-optimal. Other combinations of the graph construction method (e.g., the persistent homology [24]), distance metric (e.g., using deep metric learning), and kernel function (e.g., the Gaussian kernel) are worth exploring. Second, NRAE can be extended to a stochastic model in a number of ways, e.g., using the probabilistic autoencoder [3] that uses the latent variable model, or the energy-based approach as done in [23].

## Acknowledgments and Disclosure of Funding

This work was supported by the NAVER LABS AMBIDEX Project, NRF-2016R1A5A1938472, SNU-IAMD, SNU BK21+ Program in Mechanical Engineering, and the SNU Institute for Engineering Research.

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
