# Supplementary Material of Neighborhood Reconstructing Autoencoders

## 1 Experimental Details

In what follows we will call each experiment by its corresponding figure or table number for convenience.

### 1.1 Dataset

Except for a few synthetic data whose generation processes are described in the main script (Figure 4, 7), we use the standard benchmark datasets downloaded from TorchVision library. For the rotated/shifted MNIST images (Figure 8, 9), we use the Affine transformation function in the TorchVision library.

For Figure 5, we select the first 100 images of digit 8 from 50000 training data in the original MNIST dataset and rotate by 3.6 degrees 100 times to generate a new 10000 training data. For Figure 6, we use the entire 50000 training data for training. For the rotated MNIST of digit 3 in Figure 8, we select the first image of digit 3 from the original MNIST dataset and rotate by 9 degrees 20 times to generate a new 20 training data. For the shifted MNIST of digit 7 in Figure 8, we select the first image of digit 7 from the original MNIST dataset and transform with scale 0.8 and shift range [-10,10] to generate 20 training data. For Figure 9, we select the first image of digit 6 from the original MNIST dataset and rotate by 9 degrees 20 times to generate a new 20 training data. For Figure 10, we use the 1000,2000,...,10000 training data selected from the training dataset, 10000 validation data, and 50000 test data.

In experiments (Table 2, 3, 4, 5), we use either or both of the Large (L) and Small (S) dataset for the standard benchmark vision data: MNIST, FMNIST, KMNIST, Omniglot, SVHN, CIFAR10, CIFAR100, CELEBA. The large denotes the use of entire public data where training, validation, and test splits are (50000,10000,10000) for MNIST, FMNIST, KMNIST, (15000,4280,13180) for Omnigolot, (60000, 13257, 26032) for SVHN, (45000,5000,10000) for CIFAR10, CIFAR100, and (162770,19867,19962) for CELEBA. The small denotes the use of 20 to 30 percents of the entire public training data where training, validation, and test splits are (10000,2000,50000) for MNIST, FMNIST, KMNIST, SVHN, CIFAR10, CIFAR100, (8000,1000,4180) for Omnigolot, and (50000,10000,100000) for CELEBA. For Table 2, we use the large dataset. For Table 3, we use both large and small datasets. For Table 4 and 5, we use the small dataset.

### 1.2 Network Architecture

In this paper, we use fully connected neural network and convolutional neural network. For VAE, we use the Gaussian encoders whose output dimension is always doubled to represent both mean and variance, and use the isotropic Gaussian decoders with trainable variances. For DAE, we use the Gaussian noise in training. For WAE, we use the MMD loss and median heuristic.

**Fully connected neural network (Figure 4, 5, 6, 7, 8, 9, 10)** For Figure 4, we use the networks of size (2-1024-1024-1) and (1-1024-1024-2) with ReLU activation functions for the encoder and decoder, respectively. For Figure 5, we use the networks of size (784-1024-1024-1024-2) and (2-1024-1024-1024-784) with ReLU activation functions for the encoder and decoder, respectively. For

Figure 6, we use the networks of size (784-500-500-2) and (2-500-500-784) with Softplus activation functions (a smooth approximation of the ReLU function) for the encoder and decoder, respectively. For Figure 7, we use the networks of size (2-512-512-1) and (1-512-512-2) with ReLU activation functions for the encoder and decoder, respectively. For the rotated MNIST of digit 3 in Figure 8, we use the networks of size (784-32-32-1) and (1-32-32-784) with ReLU activation functions for the encoder and decoder, respectively. For the shifted MNIST of digit 7 in Figure 8, we use the networks of size (784-128-128-128-128-1) and (1-128-128-128-128-784) with ReLU activation functions for the encoder and decoder, respectively. For Figure 9, we use the networks of size (784-512-512-2) and (2-512-512-784) with ReLU activation functions for the encoder and decoder, respectively, where the latent values are normalized after encoding. For Figure 10, we use the networks of size (784-512-512-$m$) and ($m$-512-512-784) with ELU activation functions for the encoder and decoder, respectively. $m$ is 16, 32, and 32 for MNIST, FMNIST, and KMNIST, respectively.

**Convolutional neural network (Table 2, 3, 4, 5)**    We will denote a convolution and deconvolution layer by Conv2d(a,b,c,d,e) and ConvTranspose2d(a,b,c,d,e), respectively, where a is the number of input channel, b is the number of output channel, c is the kernel size, d is the stride (biases are always set True), and e is the padding. We use ReLU activation functions.

For MNIST, FMNIST, KMNIST, Omniglot whose image sizes are (1,28,28), we use the convolutional encoder networks

$$\text{Conv2d(1,128,3,2,0) - Conv2d(128,256,3,2,0) -}$$
$$\text{Conv2d(256,512,3,2,0) - Conv2d(512,1024,3,2,0) - Conv2d(1024,}z_{\text{dim}}\text{,1,1,0),}$$

and the decoder networks

$$\text{ConvTranspose2d(}z_{\text{dim}}\text{,1024,8,1,0) - ConvTranspose2d(1024,512,3,2,1) -}$$
$$\text{ConvTranspose2d(512,256,2,2,1) - ConvTranspose2d(256,1,1,1,0),}$$

where $z_{\text{dim}}$ is 16,32,32,32, respectively.

For SVHN, CIFAR10, CFIAR100 whose image sizes are (3,32,32), we use the convolutional encoder networks

$$\text{Conv2d(3,128,4,2,0) - Conv2d(128,256,4,2,0) -}$$
$$\text{Conv2d(256,512,4,2,0) - Conv2d(512,1024,2,2,0) - Conv2d(1024,}z_{\text{dim}}\text{,1,1,0),}$$

and the decoder networks

$$\text{ConvTranspose2d(}z_{\text{dim}}\text{,1024,8,1,0) - ConvTranspose2d(1024,512,4,2,1) -}$$
$$\text{ConvTranspose2d(512,256,4,2,1) - ConvTranspose2d(256,3,1,1,0),}$$

where $z_{\text{dim}}$ is 64,128,128, respectively.

For CELEBA whose image size is (3,64,64), we use the convolutional encoder networks

$$\text{Conv2d(3,128,5,2,0) - Conv2d(128,256,5,2,0) -}$$
$$\text{Conv2d(256,512,5,2,0) - Conv2d(512,1024,5,2,0) - Conv2d(1024,}z_{\text{dim}}\text{,1,1,0),}$$

and the decoder networks

$$\text{ConvTranspose2d(}z_{\text{dim}}\text{,1024,8,1,0) - ConvTranspose2d(1024,512,4,2,1) -}$$
$$\text{ConvTranspose2d(512,256,4,2,1) - ConvTranspose2d(256,128,4,2,1) -}$$
$$\text{ConvTranspose2d(128,3,1,1,0),}$$

where $z_{\text{dim}}$ is 128.

## 1.3   Training Details

We first introduce how we choose the hyperparameter for each algorithm and experiment. AE, VAE do not have hyperparameters, WAE, CAE, SPAE have the regularization coefficients, DAE has the noise level, and NRAE-L, NRAE-Q have $\lambda$. For NRAE, we fix the number of nearest neighbors

as 15 for MNIST, FMNIST, KMNIST, Omniglot, SVHN, CELEBA (S) and 30 for CIFAR10, CIFAR100, CELEBA (L). We note that these values are not found through extensive search processes, nevertheless, our algorithms show good results. The same numbers of nearest neighbors are used for SPAE except for the CELEBA where we choose 15 nearest neighbors because the computational complexity of SPAE largely increases as the number of nearest neighbors increases.

For experiments that show qualitative results (Figure 4, 5, 6, 7, 8, 9), we try our best for searching proper hyperparameters for all experiments. For experiments that show quantitative results, we select the best hyperparameters based on the reconstruction error metrics evaluated with the validation data. For Figure 10, Table 3, the regularization coefficients for CAE, WAE are searched around $0.01 \sim 0.001$, the noise level used in DAE is searched around $0.1 \sim 0.01$, and the regularization coefficient and $\lambda$ for SPAE and NRAE are searched around $0.001 \sim 0.0001$. For Table 2, we use the best hyperparameters selected from the Table 3, 5 except for the DAE (we use the same noise statistics for DAE in training as the added noise statistics). For Table 4, we search the parameters over the joint hyperparameter spaces.

For Figure 10 and Table 3, 4, 5, we use the Adam optimizer with learning rates $0.001 \sim 0.0001$ for 1000 epochs using 100 batch size. We use the following early-stopping criteria: we stop the training if the mean reconstruction error on validation data increases 10 times in a row. For Table 2, we did not use the early-stopping since we do not have clean data in this setting.

In experiments, we use the following GPU: TITAN X (Pascal), GeForce GRX 1080 Ti, GeForce RTX 2080 Ti, GeForce RTX 3090, each of which RAM has $10 \sim 24$ GB memory. For CAE, we need 24 GB RAM. We did not use multiple GPUs for each experiment; a single machine is enough to run the experiments.

## 2 Additional Experimental Results.

### 2.1 Computational Time

In this section, all experiments are performed on TITAN X (Pascal) with 12GB RAM. We first compare the per-epoch runtime of NRAE with the vanilla AE, VAE and the graph-based method TopoAE in Table 1. The runtime of TopoAE is adopted from the original paper, and we take experiments of AE, VAE, and NRAE with the same setting (100 batch size with MNIST dataset, 3 hidden FC layers). Although the device and other environments used in TopoAE experiment can be different from those used in our experiments, as shown in Table 1, the difference in computational time is big enough to compensate those differences in devices and environments. As discussed in the main script, TopoAE-like methods that require to compute the topological features using the persistent homology at every training iteration are yet computationally very expensive.

Table 1: Comparisons of the per-epoch runtime with 100 batch size with MNIST dataset. The network architecture is composed of 3 hidden FC layers (1000-500-250) and (250-500-1000) for the encoder and decoder, respectively, with two-dimensional latent space (The runtime of TopoAE is adopted from the original paper).

|  | AE | VAE | NRAE-L | NRAE-Q | TopoAE |
|---|---|---|---|---|---|
| time (s) | 1.74 | 2.17 | 3.13 | 4.15 | 68 |

Using the fully connected neural networks, we compute the per-epoch runtimes (100 batch size and 50000 training data) of our algorithms and the baselines (Figure 1). Firstly, we compute the per-epoch runtimes by changing the input dimension as 100, 500, 1000, 3000, 5000. The runtime of CAE rapidly increases because it uses the full Jacobian in the loss function, and, when the input dimension is beyond 1000, we couldn't run the experiments due to the GPU memory limitation. On the other hand, the runtimes of our algorithms are comparable with other existing methods. Secondly, we compare the runtimes of our algorithms with the graph-based method SPAE by changing the number of nearest neighborhood points. The runtime of the SPAE linearly increases (logarithmic in the graph) since it requires to compute the forward pass of the encoder function as many times as the number of neighborhood points during training. In contrast, our algorithms can use the batch sampling method for the neighborhood points, thus the runtimes maintain constant.

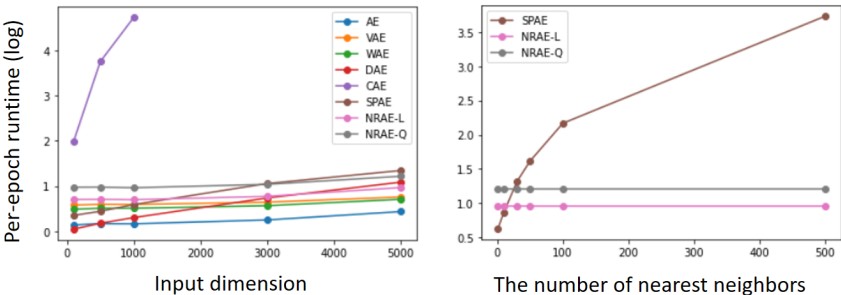

Figure 1: The per-epoch runtimes as the functions of the input dimension (left) and the number of nearest neighbors (right).

## 2.2 Robustness to the Choice of the Number of Nearest Neighbors

It is reasonable to ask how robust the NARE is to the choice of the number of nearest neighbors. We take an experiment with the MNIST dataset to show its behavior given varying number of nearest neighbors. Figure 2 shows the test mean reconstruction errors of NRAE as a function of the number of nearest neighbors compared to the other baselines. As shown in the figure, the generalization performances of NRAE are mostly better than the other baselines robustly to the number of nearest neighbors.

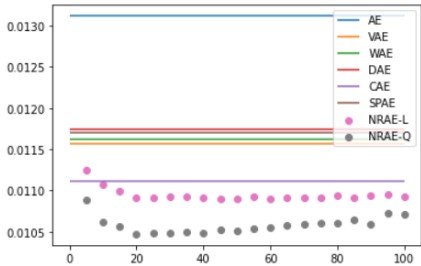

Figure 2: Comparisons of the test mean reconstruction errors of NRAE and baseline AEs. For NRAE, we report the results as a function of the number of nearest neighbors.

## 2.3 Robustness to the Choice of the Batch Size

In the main script, we set the neighborhood batch size two. We have conducted additional experiments to test the robustness of our approach to the choice of neighborhood batch size (We use MNIST data with a 16-dimensional latent space). For batch sizes (2, 4, 6, 8, 10, 12), the corresponding reconstruction losses of NRAE-L are (0.00953, 0.00952, 0.00950, 0.00952, 0.00955, 0.00945), while those of NRAE-Q are (0.00968, 0.00966, 0.00975, 0.00983, 0.00975, 0.00982), which we think is quite robust.

## 2.4 Extension of the Table 3

In Table 3 of the main script, we only report the test MSE as a measure of the generalization performance due to the space limitation. To better understand the algorithms, here we report the following additional measures: the Frechet-Inception Distance (FID) scores and the Evidence Lower Bound (ELBO). Also, to see the variances of the algorithms, we repeat the experiments with small datasets five times to compute and report the standard errors of the test MSEs.

**FID score and ELBO** While the mean reconstruction error is one of the most intuitive methods to measure the difference between data, there are other similarity measures frequently used in the community. The Frechet-Inception Distance (FID) score is a measure of similarity between two datasets of images. It was shown to correlate well with human judgement of visual quality and is most often used to evaluate the quality of generated samples. FID score is calculated by computing the

Fréchet distance between two Gaussians fitted to latent feature representations of two sets of images. We measure the FID score with a provided model from `github.com/mseitzer/pytorch-fid`.

As another way to measure the quality of the learned manifold or generalization performance, we convert the trained deterministic models to stochastic models and compare the Evidence Lower Bound (ELBO) (a lower bound of log probability) evaluated with the test data.

Once $f_\theta$ and $g_\phi$ of autoencoder are trained, adopting the idea in [1], we define a latent variable model to estimate $p_\mathcal{D}(x)$ as follows:

$$p_{\sigma,\gamma}(x) = \int_z p_\sigma(x|z)p_\gamma(z)dz, \tag{1}$$

where $p_\gamma(z)$ is a parametric density model such as the normalizing flow models [3, 4, 6, 5], and $p_\sigma(x|z)$ is the stochastic decoder defined as the Gaussian ansatz:

$$p_\sigma(x|z) := \frac{1}{\sqrt{(2\pi)^n \prod_{i=1}^n \sigma_i^2}} \exp(-\frac{1}{2}\sum_{i=1}^n \frac{(x_i - (f_\theta(z)_i))^2}{\sigma_i^2}), \tag{2}$$

where the noise covariance is the diagonal matrix with $\sigma = (\sigma_1, \ldots, \sigma_n)$. We interpret the deterministic encoder function $g_\theta$ as a stochastic encoder $q_\epsilon(z|x) = \mathcal{N}(g_\theta(x), \epsilon^2\mathbb{I})$ with a learnable scalar parameter $\epsilon$, and train $p_{\sigma,\gamma}$ and $q_\epsilon(z|x)$ by maximizing the evidence lower bound (ELBO) as in variational autoencoders training [7] ($\theta, \phi$ are fixed during training), where the ELBO at data point $x$ is obtained as follows:

$$\text{ELBO}(x) = \mathbb{E}_{q_\epsilon(z|x)}[\log p_\sigma(x|z)] - D_{\text{KL}}(q_\epsilon(z|x)||p_\gamma(z)), \tag{3}$$

where $D_{\text{KL}}$ is the KL-divergence. We use the realnvp model [4] for $p_\gamma$ where the depth is 8, the lengths of hidden vectors are 32, and resale and permutation are set true.

**Test Reconstruction MSEs, FID scores, and ELBOs.**    As shown in Table 2, the ELBOs are likely to be high if the MSEs are low because the learned density functions tend to assign high probability densities to the test data whose reconstruction errors are low. The NRAE-L and NRAE-Q mostly show lower MSEs and higher ELBOs than the other baselines. However, for some cases such as the FMNIST, KMNIST, and Omniglot, the CAE or SPAE produce higher ELBOs even though their MSEs are higher than the NRAEs. This is because the ELBOs not only depend on the quality of the learned manifolds but also how easily their encoded latent space distributions can be fit with the normalizing flow models $p_\gamma$ (i.e., minimizing the KL-divergence term). The MSE and ELBO results in Table 2 imply that the NRAE-L and NRAE-Q i) mostly learn better manifolds (i.e. low MSEs) than the other baselines yet sometimes ii) produce latent space distributions that are difficult to be learned with $p_\gamma$. Although studying how to train the autoencoder in a way that its latent space distribution can be easily learned is an out-of-scope of this paper, developing a new regularization method that can be added to the NRAE for making its latent space distribution easier to be learned would be an interesting future direction.

On the other hand, the FID scores are not always positively correlated to the MSEs. Since our algorithms are implemented using the Euclidean distance metric for graph construction, the FID scores may not be lower than the others. Nevertheless, for some examples especially small datasets, the NRAE-L and NRAE-Q produce not only lower MSEs but also lower FID scores. For example, the NRAE-L and NRAE-Q show lower MSEs and FID scores for SVHN (S,L), CIFAR10 (S), and CIFAR100 (S), but show lower MSEs yet higher FID scores for CIFAR10 (L), CELEBA (L).

Figure 3 shows some of the test image reconstruction results of SVHN (S) and CIFAR10 (S). Not only the MSEs and FID scores are lower, but also visual qualities of the reconstructed images by the NARE-L and NRAE-Q are much better than the other baselines. Figure 4 shows some of the test image reconstruction results of CIFAR10 (L) and CELEBA (L). For CIFAR10 (L), reconstructed results are not significantly different, visually. For CELEBA (L), where the NRAE-L and NRAE-Q show the lowest MSEs but the highest FID scores, reconstructed results of the NRAEs are little more blurry than the others. Our algorithms seem to overly smooth out the CELEBA data. We believe that this can be alleviated either by decreasing the number of nearest neighbors or decreasing $\lambda$. Although we didn't have enough time to empirically prove that NRAEs can better perform on CELEBA (L) in FID scores, in principle, there exist proper number of nearest neighbors and $\lambda$ with

which NRAEs perform at least better or equal to the vanilla AE in FID scores, because the NRAEs have the convergence (to the vanilla AE) property.

It is remarkable that, especially for SVHN, CIFAR10, CIFAR100, CELEBA datasets, the NRAE-L and NRAE-Q trained with small datasets i) largely outperform the other baselines trained with small datasets and ii) show comparable performances compared to the other baselines trained with large datasets. This shows that our algorithm, by leveraging the local geometric information contained in the neighborhood graph, has a significant advantage in generalization when the number of training data is small.

Table 2: The test reconstruction MSEs, FID scores (the lower the better) and ELBO (the higher the better). The FID scores are computed on RGB-image datasets only. The best and second-best are colored red and blue, respectively.

| Dataset | Metric | Size | AE | VAE | WAE | DAE | CAE | SPAE | NRAE-L | NRAE-Q |
|---|---|---|---|---|---|---|---|---|---|---|
| MNIST | MSE | S | 0.01002 | 0.01091 | 0.01009 | 0.00999 | 0.00998 | 0.00989 | 0.00953 | 0.00968 |
| | | L | 0.00688 | 0.00756 | 0.00690 | 0.00684 | 0.00692 | 0.00694 | 0.00649 | 0.00683 |
| | ELBO | S | 202.59 | 209.31 | 208.34 | 201.38 | 252.31 | 239.96 | 371.01 | 413.16 |
| | | L | 327.65 | 223.62 | 375.6 | 366.38 | 458.34 | 375.62 | 631.17 | 658.87 |
| FMNIST | MSE | S | 0.01485 | 0.01652 | 0.01428 | 0.01446 | 0.01319 | 0.01363 | 0.01289 | 0.01277 |
| | | L | 0.01118 | 0.01235 | 0.01106 | 0.01099 | 0.01052 | 0.01065 | 0.01060 | 0.01044 |
| | ELBO | S | 311.30 | 285.37 | 347.14 | 320.51 | 494.85 | 421.59 | 410.97 | 430.03 |
| | | L | 434.51 | 398.10 | 443.26 | 443.44 | 580.62 | 533.97 | 498.14 | 505.10 |
| KMNIST | MSE | S | 0.03267 | 0.03234 | 0.03283 | 0.03280 | 0.03279 | 0.03268 | 0.03071 | 0.03021 |
| | | L | 0.02844 | 0.02963 | 0.02776 | 0.02814 | 0.02762 | 0.02732 | 0.02564 | 0.02602 |
| | ELBO | S | -19.19 | 22.17 | 12.98 | -23.78 | 96.98 | 63.03 | 42.89 | 59.58 |
| | | L | 35.96 | 43.65 | 66.34 | 42.58 | 174.05 | 131.77 | 112.32 | 120.35 |
| Omniglot | MSE | S | 0.03038 | 0.03627 | 0.03078 | 0.03068 | 0.02714 | 0.02889 | 0.02668 | 0.02631 |
| | | L | 0.02704 | 0.03192 | 0.02728 | 0.02696 | 0.02567 | 0.02644 | 0.02578 | 0.02539 |
| | ELBO | S | 33.00 | -24.62 | 35.93 | 30.60 | 150.62 | 110.65 | 96.96 | 117.34 |
| | | L | 92.10 | 20.78 | 90.65 | 97.89 | 189.11 | 141.95 | 132.96 | 148.22 |
| SVHN | MSE | S | 0.00320 | 0.00420 | 0.00320 | 0.00369 | 0.00273 | 0.00307 | 0.00202 | 0.00192 |
| | | L | 0.00174 | 0.00204 | 0.00190 | 0.00177 | 0.00178 | 0.00175 | 0.00148 | 0.00147 |
| | ELBO | S | 1146.20 | 987.80 | 1015.34 | 1145.97 | 1304.49 | 1291.12 | 3908.37 | 4050.09 |
| | | L | 3576.44 | 3307.51 | 3100.47 | 3567.15 | 4330.74 | 4130.41 | 5134.48 | 4762.67 |
| | FID | S | 91.69 | 124.51 | 105.00 | 90.54 | 60.48 | 77.38 | 31.54 | 28.61 |
| | | L | 40.44 | 40.88 | 40.16 | 38.34 | 41.02 | 40.20 | 36.95 | 35.95 |
| CIFAR10 | MSE | S | 0.01466 | 0.01620 | 0.01431 | 0.01427 | 0.01208 | 0.01504 | 0.00768 | 0.00691 |
| | | L | 0.00960 | 0.01123 | 0.00863 | 0.00900 | 0.00755 | 0.00898 | 0.00629 | 0.00587 |
| | ELBO | S | 565.04 | 269.54 | 398.32 | 547.34 | 908.45 | 768.79 | 1963.12 | 1823.59 |
| | | L | 520.46 | 342.21 | 425.80 | 631.54 | 1813.96 | 930.47 | 2607.25 | 2643.33 |
| | FID | S | 137.12 | 157.18 | 132.79 | 133.82 | 108.39 | 122.20 | 94.27 | 85.73 |
| | | L | 77.43 | 94.43 | 71.51 | 74.91 | 62.14 | 66.05 | 68.74 | 70.53 |
| CIFAR100 | MSE | S | 0.01465 | 0.01713 | 0.01463 | 0.01484 | 0.01369 | 0.01477 | 0.00765 | 0.00717 |
| | | L | 0.01015 | 0.01064 | 0.00951 | 0.00862 | 0.00842 | 0.00912 | 0.00678 | 0.00635 |
| | ELBO | S | 571.81 | 257.07 | 522.22 | 508.72 | 812.30 | 772.32 | 2004.62 | 1603.06 |
| | | L | 625.13 | 354.69 | 481.51 | 468.68 | 1446.45 | 917.11 | 2463.61 | 2498.37 |
| | FID | S | 122.28 | 139.20 | 131.91 | 156.50 | 127.29 | 108.86 | 85.56 | 80.06 |
| | | L | 81.35 | 86.96 | 78.77 | 73.95 | 65.02 | 66.68 | 64.84 | 68.02 |
| CELEBA | MSE | S | 0.00780 | 0.00937 | 0.00830 | 0.00782 | - | 0.00861 | 0.00608 | 0.00747 |
| | | L | 0.00613 | 0.00646 | 0.00630 | 0.00590 | - | 0.00665 | 0.00563 | 0.00565 |
| | ELBO | S | 4298.56 | 4101.55 | 6043.03 | 4311.07 | - | 4513.48 | 12934.30 | 11692.26 |
| | | L | 11224.41 | 10628.09 | 11146.31 | 11634.83 | - | 11328.64 | 13456.26 | 13457.03 |
| | FID | S | 60.40 | 70.13 | 60.40 | 59.92 | - | 60.15 | 57.51 | 68.59 |
| | | L | 43.02 | 45.18 | 44.74 | 42.55 | - | 43.54 | 57.70 | 55.28 |

**Standard errors of MSEs (5 times run).** In this section, we report the means and standard errors for 5 times run of NRAE and baseline AEs for small datasets. The hyperparameters during 5 times run for each AE are same with the settings whose results are reported in Table 3 of Section 4.3. In table 3, we report the means and standard errors of the test reconstruction MSEs. As shown in the table, the standard errors are negligible and the NRAE-L and NRAE-Q show lower MSEs than the other baselines.

## 3    Further Discussion

**Relation to local polynomial regression.** Smoothing data points by fitting a local polynomial model is a well-known technique in non-parametric regression [2]. Assuming a fixed encoder function $g$, training the decoder in NRAE shares some similarities to local polynomial regression. For example, given a set of paired data $\mathcal{D}_p := \{(z, x) | z = g(x), x \in \mathcal{D}\}$, the local linear regression

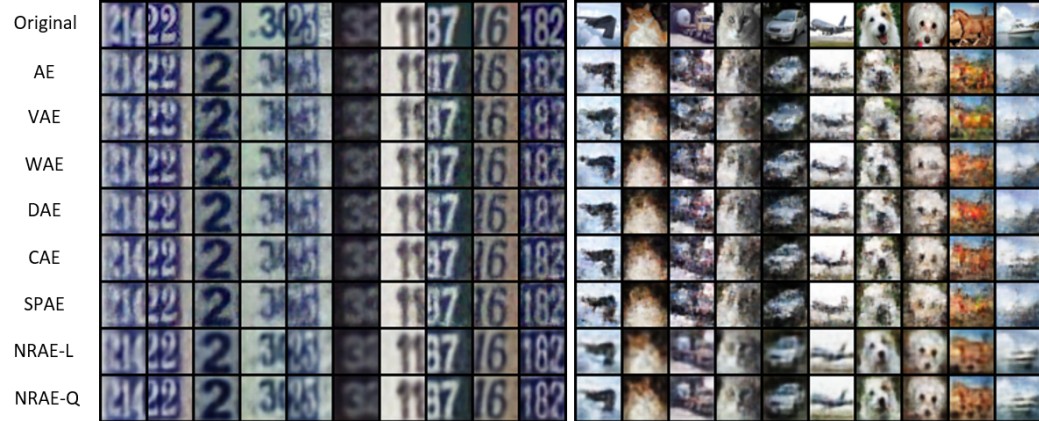

Figure 3: The test image data reconstruction results where the NRAE-L and NRAE-Q show lower MSEs and FID scores than the other baselines. (left) SVHN (S), (right) CIFAR10 (S).

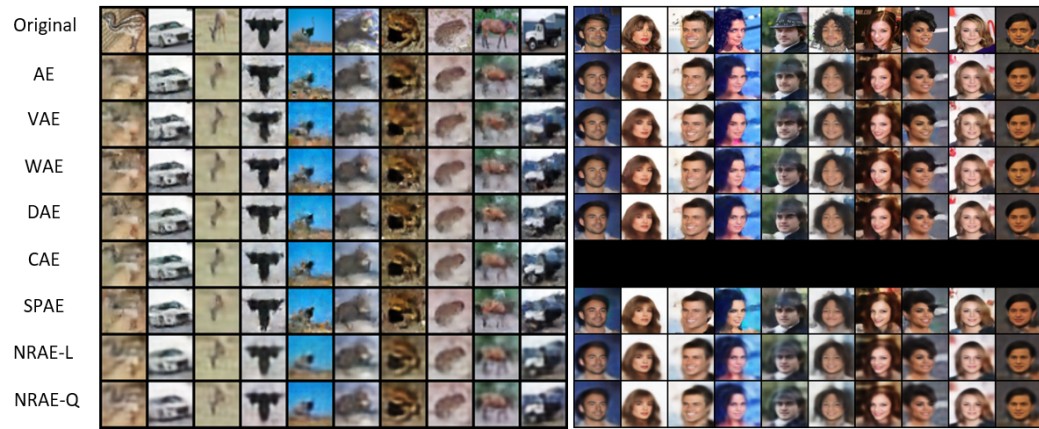

Figure 4: The test image data reconstruction results where the NRAE-L and NRAE-Q show lower MSEs but higher FID scores than the other baselines. (left) CIFAR (L), (right) CELEBA (L).

problem for estimating $f$ at $z$ is typically formulated as follows:

$$f(z), A^*(z) = \underset{x \in \mathbb{R}^n, A \in \mathbb{R}^{n \times m}}{\operatorname{argmin}} \sum_{(z', x') \in \mathcal{D}_p} K(z', z) \cdot \|x' - (x + A(z' - z))\|^2, \qquad (4)$$

where $K(z', z)$ is a kernel function, $f(z)$ is the estimate of $x$ at $z$, and $A^*(z) \in \mathbb{R}^{n \times m}$ is the estimated linear coefficient at $z$. While local polynomial regression is a non-parametric technique that requires solving an optimization for every query point $z$, NRAE learns a parametric model $f_\theta$ for a similar-looking loss function that uses a local polynomial approximation of $f_\theta$.

**Convergence to vanilla AE.** NRAE is a generalization of a vanilla AE in the following sense: NRAE converges to the vanilla AE – that is, the neighborhood reconstruction loss converges to the point reconstruction loss – if $\mathcal{N}(x) \to \{x\}$ or $K(x', x) \to \delta(x', x)$.

**Linear decoder function.** As a special case, consider an autoencoder with a linear decoder function such that $f_\theta(z) = \theta_1 z + \theta_0$ for $\theta_1 \in \mathbb{R}^{n \times m}, \theta_0 \in \mathbb{R}^n$. The second-order derivative of $f_\theta$ is zero while the first-order derivative $\frac{\partial f_\theta(z)}{\partial z} = \theta_1$; the neighborhood reconstruction loss for a point $x' \in \mathcal{N}(x)$ then becomes

$$\|x' - \tilde{F}_{\theta,\phi}(x'; x)\|^2 := \|x' - \theta_1 g_\phi(x) - \theta_0 - \theta_1(g_\phi(x') - g_\phi(x))\|^2 = \|x' - F_{\theta,\phi}(x')\|^2. \quad (5)$$

The above implies that NRAE with a linear decoder function is identical to the vanilla AE with the linear decoder.

Table 3: The means and standard errors of the test reconstruction MSEs (the lower the better). The metrics are computed with 5 times run except that metrics on the CELEBA data are computed with 3 times run. The best and second-best results are colored red and blue, respectively.

| Dataset | Statistic | AE | VAE | WAE | DAE | CAE | SPAE | NRAE-L | NRAE-Q |
|---|---|---|---|---|---|---|---|---|---|
| MNIST | mean | 0.010670 | 0.01094 | 0.01075 | 0.01065 | 0.01037 | 0.01064 | 0.00971 | 0.01013 |
| | ste | ±0.00028 | ±0.00006 | ±0.00031 | ±0.00026 | ±0.00023 | ±0.00034 | ±0.00018 | ±0.00020 |
| FMNIST | mean | 0.01435 | 0.01656 | 0.01391 | 0.01403 | 0.01303 | 0.01342 | 0.01281 | 0.01273 |
| | ste | ±0.00015 | ±0.00003 | ±0.00013 | ±0.00014 | ±0.00008 | ±0.00009 | ±0.00003 | ±0.00003 |
| KMNIST | mean | 0.03254 | 0.03251 | 0.03306 | 0.03283 | 0.03213 | 0.03255 | 0.03053 | 0.02996 |
| | ste | ±0.00008 | ±0.00008 | ±0.00013 | ±0.00027 | ±0.00026 | ±0.00019 | ±0.00012 | ±0.00010 |
| Omniglot | mean | 0.03028 | 0.03146 | 0.03155 | 0.03086 | 0.02891 | 0.02886 | 0.02684 | 0.02631 |
| | ste | ±0.00018 | ±0.00109 | ±0.00109 | ±0.00020 | ±0.00040 | ±0.00011 | ±0.00006 | ±0.00016 |
| SVHN | mean | 0.00323 | 0.00431 | 0.00319 | 0.00333 | 0.00271 | 0.00310 | 0.00205 | 0.00220 |
| | ste | ±0.00002 | ±0.00012 | ±0.00004 | ±0.00009 | ±0.00004 | ±0.00005 | ±0.00002 | ±0.00019 |
| CIFAR10 | mean | 0.01472 | 0.01709 | 0.01445 | 0.01486 | 0.01251 | 0.01465 | 0.00816 | 0.00700 |
| | ste | ±0.00032 | ±0.00026 | ±0.00022 | ±0.00036 | ±0.00011 | ±0.00015 | ±0.00070 | ±0.00006 |
| CIFAR100 | mean | 0.01463 | 0.017318 | 0.014548 | 0.01435 | 0.01361 | 0.01455 | 0.00781 | 0.00732 |
| | ste | ±0.00008 | ±0.00019 | ±0.00023 | ±0.00022 | ±0.00026 | ±0.00017 | ±0.00011 | ±0.00005 |
| CELEBA | mean | 0.00797 | 0.00914 | 0.00821 | 0.00780 | - | 0.00839 | 0.00602 | 0.00727 |
| | ste | ±0.00012 | ±0.00010 | ±0.00009 | ±0.00001 | - | ±0.00009 | ±0.00003 | ±0.00008 |