# OpenReview forum: "Neighborhood Reconstructing Autoencoders"
_NeurIPS.cc/2021/Conference — NeurIPS 2021 Poster_

### Official Review · Reviewer_jN5d · 2021-06-30

**Rating:** 6
**Confidence:** 4

**Summary:**

An auto-encoding (AE) loss that reconstructs data according to a local approximation of the autoencoding function. In particular, for each $x_i$ in the dataset, the model reconstructs data around a neighborhood of $x_i$ using a low-order Taylor expansion of the encoder-decoder composition around $x_i$. The loss to each point in the neighborhood is assigned a weight according to a simple kernel function. The author demonstrates empirical advantage compared to many other AE losses in extensive evaluations.

**Limitations And Societal Impact:**

The very last section of limitations did not really talk about limitations but mostly future works and extensions. Please consider my comments above and refine. An ideal work should address theory, explain why a version of the algorithm can avoid problems of another version (linear vs quadratic).

**Main Review:**

# Originality
I am familiar with AE but not with tree-based AE methods. I find the idea of this work novel interesting. In particular, the local approximation of the full encoder and decoder composition seems novel.

# Quality
This work appears to be very much intuition-based and easy to understand. The authors also provided key visualizations of their method at work. My main concerns are
1. the discussion on the source of overfitting is confusing, and I do not very much follow the argument. Although I see it is connected with the later proposal of approximating the encoder-decoder composition.
2. a lack of theoretical analysis
3. The toy experiments may be quite unfair to traditional methods, and a think the authors should either emphasize or remind readers of the assumptions made or run more experiments in a fairer comparison. In particular, for datasets with very few data points, I think the comparison with many other AEs is a bit unfair due to finite data. The problem of failing to denoise should go away if using more data.
4. Please provide intuition why NRAE-Q can be better or worse compared with NRAE-L, especially in the context of Figure 7.

# Clarity
Very clear and easy to understand, although some visualizations can be improved to avoid confusion at first sight. Details to follow.

# Significance
Despite the lack of theoretical components, I find this work to be interesting and the empirical validations thorough, although the toy data raises some questions that I hope to see answers to from the authors.

# Detailed comments

Detailed comments:

a. Line 37-38: I don't see whether Figure 1(a) provides the intuition for the argument in this line. It could be that the encoder does not prune the noise (if z is 2-d), giving overfitting. How does it show the distinction of overfittings caused by encoder or decoder?

b. Line 53-63: I found these detail to be repetitive of what follows, and I spent quite a while understanding these, and then realize that the details in Section 2 are much more clear.

c. Line 87: $\hat{F}$ of a set $\mathcal{N}$ is not defined, but I guess it maps to a set through element-wise operation?

d. Figure 2 is very confusing but can be very useful if the readers are reminded that the start blue point is on the short spiky part of the data manifold in the second panel, and thus the red points seem to be crossing the apparent data manifold. I see the pale orange model manifold in the fourth panel as well, but the important spike was pretty much covered by the red dots. I was confused for quite a while.

e. Line 123: Please provide more details for the batch sampling. For each minibatch $B = [x_1, x_2,...,x_k]$, do you sample a neighbour for each of the $k$ samples? Does B_x only contain two samples? B_x doesn't really appear afterward.

f. Line 132.  I find (5) a bit confusing. What is the domain of optimization? Is $x\in\mathcal{D}$ fixed? but somehow it is optimized over? I'm not familiar with the original literature so excuse me if this should be obvious.

g. Line 139. What is the notion of convergence to a set with countable elements? Better replace with =?

h. Line 144. It is identical to the vanilla AE *with linear decoder*?

i. Section 4.1, second experiment, how do you normalize each latent space? I'm not sure I understood the last sentence about DAE.

j. Line 212, Table 4.1 -> Table 1

k. Figure 7. The problem of the failing cases seems to be that there is a spurious manifold jumping from the bottom point circled red to the inner tip of the spiral, spiraling around to the other point at the bottom and then tracing back to the inner tip. Is it just a consequence of the bigger gap between the points in circles? Again, this should not happen if the dataset is larger.  I really want to see why NRAE-Q avoided this problem since it seems that with a quadratic approximation this problem can happen more easily.

**Time Spent Reviewing:**

2

---

> ### Author Response · Authors · 2021-08-10
> **Response for Reviewer jN5d**
>
> Dear Reviewer jN5d,
>
> Thank you very much for your informative and constructive review. Your comments will certainly improve the manuscript.
>
> **Q1.** The discussion on the source of overfitting is confusing, and
> I do not very much follow the argument. Although I see it is connected with
> the later proposal of approximating the encoder-decoder composition.
>
> **A1.** In retrospect, it appears that some of our original statements
> were insufficiently precise and may have led to unintended implications (see
> some of our responses to the other reviewers, which we will also repeat here).
>
> To clarify, both the encoder and decoder can contribute to overfitting. A more
> accurate statement of our assertion would therefore be that encoder-only
> regularization methods alone are insufficient to mitigate overfitting; we
> take into account both the encoder and decoder, and through our analysis
> arrive at the conclusion that "matching the local geometry of the decoded manifold to that of the original data distribution significantly improves the reconstruction quality (i.e., learning the correct manifold)".
>
> **Q2.** a lack of theoretical analysis
>
> **A2.**  Theoretical analysis can take many forms, and like everyone else
> we struggled with finding the right balance between clarity of exposition of
> the central idea and theoretical rigor and completeness.  We did consider
> strengthening the analysis of convergence for training, neighborhood reconstruction loss, etc. One reviewer also asked the assumption of a smooth
> manifold was implicit in our method (it is), and what if any modifications would
> be needed in the event that, e.g., the manifold had boundaries or self-intersections,
> or consisted of multiple manifolds of different dimensions and topologies (our answer was a complete discussion of these myriad issues would require numerous assumptions and engineering choices that would detract from the main message
> of our paper). In any event, it would be very helpful to us if you could
> let us know what specific theoretical analysis you felt was important.
>
> **Q3.** The toy experiments may be quite unfair to traditional methods,
> and a think the authors should either emphasize or remind readers of the
> assumptions made or run more experiments in a fairer comparison. In particular,
> for datasets with very few data points, I think the comparison with many
> other AEs is a bit unfair due to finite data. The problem of failing to
> denoise should go away if using more data.
>
> **A3.** We do believe that our experiments are fair, as we took great care
> to ensure that all experiments were conducted under identical settings. For the
> so-called toy experiments, we repeatedly observe the same qualitative results,
> which says something. That being said, we do agree that more experiments
> will be helpful; we are conducting more experiments and will report
> the mean statistics of the test set MSE in the updated manuscript.
>
> The reviewer's comment made us realize that for the toy experiment in Figure 4,
> we did not explicitly state what latent space dimension is used --  we use a
> one-dimensional latent space, which can be inferred from the observation that
> the learned manifolds are depicted as 1-d curves in the relevant figures.
> We will add this clarification in the updated manuscript.
>
> We also agree with the reviewer's point that the problem of failing to
> denoise goes away given sufficiently many data. That being said, there are
> many real-life problems and application scenarios in which the amount of data
> is simply insufficient, and for such problems, our approach is clearly relevant.
>
> **Q4.** Please provide intuition why NRAE-Q can be better or worse
> compared with NRAE-L, especially in the context of Figure 7.
>
> **A4.** This is a very good question, thank you. We believe that
> NRAE-Q outperforms NRAE-L when data are sparsely distributed near
> highly curved regions of the ground-truth manifold.
> For example, in Figure 7, the region around the red-circled point has high
> curvature and locally sparse data distribution; here the
> NRAE-L fails to learn the correct local geometry.  More specifically,
> the linear approximation of the manifold at $x$ becomes
> less accurate as one deviates further from $x$.  In particular, if
> the curvature (more precisely, the Gaussian curvature) of the
> manifold at $x$ is large, then the accuracy of the linear approximation
> decreases more quickly away from $x$.  Since the accuracy of the
> quadratic approximation of the manifold decreases more slowly than that
> of the linear approximation, using the quadratic approximation leads
> to better results as expected.
>
>
>
> **Q5.** Line 37-38: I don't see whether Figure 1(a) provides the
> intuition for the argument in this line. It could be that the encoder
> does not prune the noise (if z is 2-d), giving overfitting. How does
> it show the distinction of overfittings caused by encoder or decoder?
>
>
>
> **A5.** In Figure 1 (a), we assume that the data space is 2-d and
> the latent space is 1-d.  The latent space dimension can be inferred
> from the figure, in which the orange learned manifolds are described as
> 1-d curves.  We will clarify this point in the updated manuscript.
> In this case, the decoder needs to fold the 1-d latent space into the
> 2-d data space in a way that perfectly reconstructs the given data points,
> and naturally there are many ways to achieve this.  The example described
> in Figure 1 (a) is a result of the decoder folding the 1-d latent space
> excessively.
>
>
>
> **Q6.** Line 87: $\tilde{F}$ of a set $\mathcal{N}(x)$ is not defined, but
> I guess it maps to a set through element-wise operation?
>
>
> **A6.** Thank you for pointing this out, you are right; We will clarify
> this point in the updated manuscript.
>
>
>
> **Q7.** Line 123: Please provide more details for the batch sampling.
> For each minibatch, do you sample a neighbour for each of the samples? Does
> $\mathcal{B}_x$ only contain two samples? $\mathcal{B}_x$ doesn't really appear
> afterward.
>
>
> **A7.** Yes, we sample a batch $\mathcal{B}_x$ for all $x\in\mathcal{B}$.
> And yes, $\mathcal{B}_x$ contains two samples.  We use $\mathcal{B}$ and
> $\mathcal{B}_x$ to compute the stochastic gradient of the objective function in
> Equation (3).
>
>
>
> **Q8.** Line 132. I find (5) a bit confusing. What is the domain of
> optimization? Is  fixed? but somehow it is optimized over? I'm not familiar
> with the original literature so excuse me if this should be obvious.
>
>
>
> **A8.** The domain is fixed as $\mathbb{R}^{n}\times \mathbb{R}^{n\times m}$.
> We will clarify this point in the updated manuscript.
>
>
>
> **Q9.** Line 139. What is the notion of convergence to a set with countable
> elements? Better replace with =?
>
>
>
> **A9.** Yes, we use the notation under the discrete topology of a countable
> set. If you still find the symbol confusing, please let us know and we will
> modify the paper accordingly.
>
>
>
> **Q10.** Line 144. It is identical to the vanilla AE with linear decoder?
>
>
>
> **A10.** Technically, yes, thanks for pointing out. We will clarify this
> point in the updated manuscript.
>
>
> **Q11.** Section 4.1, second experiment, how do you normalize each latent
> space? I'm not sure I understood the last sentence about DAE.
>
>
>
> **A11.** We normalize each latent space by $z\mapsto \frac{z-\mu}{\sigma}$
> where $\mu$ and $\sigma$ are respectively the mean and standard deviation.
> When applying DAE, we need to select a set of ``noise statistics" to be used
> in DAE traning. In reality, we cannot know a priori the actual noise statistics,
> but in de-noising experiments (Table 1) we know exactly what noise is used
> (we add Gaussian noise to images). We use the exact same noise for DAE --
> this is clearly favorable for DAE training purposes, which is why we conclude
> that DAE takes third place. Suprisingly, even with this favorable
> setting for DAE, both NRAEs outperform DAE.
>
>
>
> **Q12.** Line 212, Table 4.1 -> Table 1
>
>
>
> **A12.** Thank you very much for pointing out this error.
>
>
>
> **Q13.** Figure 7. The problem of the failing cases seems to be that
> there is a spurious manifold jumping from the bottom point circled red to
> the inner tip of the spiral, spiraling around to the other point at the bottom
> and then tracing back to the inner tip. Is it just a consequence of the bigger
> gap between the points in circles? Again, this should not happen if the dataset
> is larger. I really want to see why NRAE-Q avoided this problem since it seems
> that with a quadratic approximation this problem can happen more easily.
>
>
>
> **A13.** I believe we've answered this question in (**A4**).
>
> We thank you again for your efforts in reviewing our paper!
>
> Best regards, Authors.

---

> > ### Comment · Reviewer_jN5d · 2021-08-22
> > **Q8**
> >
> > I thank the authors for clarifying most of my questions. Just one point regarding Q8, again on notation of Eqn (5). It estimates some sort of $\hat{x}$ right? The argmin returns $f(z)$, but the loss doesn't explicitly gives a function $f$. I think this is better clarified.

---

> > > ### Author Response · Authors · 2021-08-23
> > > **Response for Reviewer jN5d**
> > >
> > > Yes. By solving the optimization problem, we can not obtain an explicit functional form of $f$ as you said. Instead, it estimates a vector value $\hat{x}$ given $z$, i.e., $\hat{x}(z)$. We will change $f(z) \mapsto \hat{x}(z)$ to clarify that $\hat{x}(z)\in\mathbb{R}^{n}$ is the estimate of $x$ at $z$ (and not a functional form)! Thank you for the comment!

---

### Official Review · Reviewer_AQVH · 2021-07-16

**Rating:** 8
**Confidence:** 4

**Summary:**

Rather than reconstructing a given input $x$, the first- or second- order* Taylor approximations (centered at $x$) of the decoder are optimized to accurately reconstruct some neighborhood $\mathcal{N}(x)$ of $x$. More formally, the loss for a single $x, x'$ pair, where $x' \in \mathcal{N}(x)$ is written as
$K(x, x') \cdot \left \Vert x - \tilde{f}(g(x')) \right \Vert^2$
where $g$ is the encoder and $\tilde{f}(x')$ is a Taylor approximation of the decoder $f$, centered at $x$. The authors evaluate the method across a wide array of empirical experiments and show that it produces superior autoencoding performance by many measures.

*There are two variants of the algorithm -- one that uses a first-order Taylor approximation, and one that uses a second-order Taylor approximation.

**Limitations And Societal Impact:**

I have some concerns over the robustness of the hyperparameters of the model added by the neighborhood reconstruction terms. I note that the supplemental material explores the robustness of the work to the number of nearest neighbors. Have the authors also explored the robustness of the method to the kernel hyperparameter $\lambda$? Or the neighborhood batch size, which I understand has been set to 2?

**Main Review:**

The paper is clear and easy to read --- this owes as much to the elegance of the idea as it does to the writing of the authors. Additionally, the method is both well-motivated, and well-validated in synthetic and real-world experiments. Neighborhood reconstruction via local Taylor approximations is charming in its simplicity, yet clearly quite far-reaching in the way it shapes the latent embedding.

To my knowledge, such an approach has not been previously made. I find the idea both original and worthy of discussion at this conference.

**Time Spent Reviewing:**

7

---

> ### Author Response · Authors · 2021-08-10
> **Response for Reviewer AQVH**
>
> Dear Reviewer AQVH,
>
> Thank you for appreciating the value of our research.
>
> **Q1.** I have some concerns over the robustness of the hyperparameters of the model added by the neighborhood reconstruction terms. I note that the supplemental material explores the robustness of the work to the number of nearest neighbors. Have the authors also explored the robustness of the method to the kernel hyperparameter? Or the neighborhood batch size, which I understand has been set to 2?
>
> **A1.** Yes, we have explored both of these points. In the case of the former, the kernel parameter $\lambda$ acts as a regularization coefficient similar to other regularization methods, e.g., the noise scale in DAE. The larger the value of $\lambda$, the smoother the learned manifold.
>
> For the latter, we set the neighborhood batch size to two. We have conducted additional experiments to test the robustness of our approach to the choice of neighborhood batch size (We use MNIST data with a 16-dimensional latent space).  For batch sizes (2, 4, 6, 8, 10, 12), the corresponding reconstruction losses of NRAE-L are (0.00953, 0.00952, 0.00950, 0.00952, 0.00955, 0.00945), while those of NRAE-Q are (0.00968, 0.00966, 0.00975, 0.00983, 0.00975, 0.00982), which we think is quite robust.  We plan to add these experimental results to the updated manuscript.
>
> We thank you again for your efforts in reviewing our paper!
>
> Best regards, Authors.

---

### Official Review · Reviewer_JTcy · 2021-07-16

**Rating:** 3
**Confidence:** 4

**Summary:**

The paper goes into detail about two major obstacles that frequent Autoencoders, namely overfitting to training data, and learning the correct local geometry of the data. They then claim that all existing methods struggle with one or the other, or they are so computationally taxing that they aren't scalable for most problems. The paper's main contribution is that they provide a new training method/loss function that uses a local 2nd order approximation of the decoder function. The paper claims that the decoder is more important than the encoder in correctly learning the local geometry of the data. They then go into the algorithm's details and then explore the related work of regularization of autoencoders. Finally, they run experiments to demonstrate the algorithm's properties as well as show that NRAE is compatible with other methods (VAE, WAE, CAE, and SPAE) and often gets better results combining methods.

**Limitations And Societal Impact:**

Yes

**Main Review:**

The approach appears to be original and unique. However I have concerns about some of the claims in the paper.

First, the authors are overly pessimistic on the scalability of methods such as GRAE. By my estimation, both GRAE and NRAE should have the same asymptotic performance computationally, although the constants may differ. Thus the authors should compare to GRAE in their experiments as well.

The authors also do not compare to SPAE in Figure 9. This should be corrected as in Figure 8, SPAE was the only other method that correctly learned the manifold.

The authors also claim that "it is in fact the decoder, and not the encoder, that has a greater impact on learning the correct manifold". The authors do not sufficiently justify this. They show that their approach seems to work well, but that is not sufficient justification for their claim. For one thing, regularizing the bottleneck has a direct effect on the decoder as well as it changes the inputs to it. Thus it is incorrect to claim that any method that only regularizes the bottleneck is ignoring the decoder. The authors should do more experiments demonstrating very clearly that focusing on the output of the decoder is superior to other methods that perform regularization on other parts of the autoencoder.

**Time Spent Reviewing:**

8

---

> ### Author Response · Authors · 2021-08-11
> **Response for Reviewer JTcy**
>
> Dear Reviewer JTcy,
>
> Thank you for your constructive feedback. We will revise our manuscript according to your comments. We believe it will certainly improve the manuscript.
>
> **Q1.** First, the authors are overly pessimistic on the scalability
> of methods such as GRAE. By my estimation, both GRAE and NRAE should have
> the same asymptotic performance computationally, although the constants
> may differ. Thus the authors should compare to GRAE in their experiments
> as well.
>
> **A1.**  The reason for not comparing our method against GRAE was because the GRAE has not been tested with high-dimensional latent space cases in the original paper. In retrospect we agree that GRAE does merit a comparison; we have implemented GRAE and are currently performing additional comparison experiments. Some preliminary results obtained so far are summarized in the table below:
>
>
> |  Dataset  | Size | AE   |   VAE   |   WAE   | DAE | CAE | GRAE | SPAE | NRAE-L | NRAE-Q |
> |:------------:|:------:|:-------:|:-------:|:-------:|:---:|:---:|:----:|:----:|:------:|:------:|
> |  MNIST |   S  | 0.01002 |  0.01091 | 0.01009 | 0.00999 | 0.00998 | 0.01004 | 0.00989 | 0.00953  | 0.00968 |
> |              |   L  | 0.00688 |  0.00756 | 0.00690 | 0.00684 | 0.00692 | 0.00696 | 0.00694 | 0.00649  | 0.00683 |
> |  FMNIST   |   S  | 0.01485 | 0.01652 | 0.01428 | 0.01446 | 0.01319 | 0.01331 | 0.01363 | 0.01289 | 0.01277 |
> |              |   L  | 0.01118 | 0.01235 | 0.01106 | 0.01099 | 0.01052 | 0.01060 | 0.01065 | 0.01060 | 0.01044 |
> |  KMNIST   |   S  | 0.03267 | 0.03234 | 0.03283 | 0.03280 | 0.03279 | 0.03206 | 0.03268 | 0.03071 | 0.03021 |
> |              |   L  | 0.02844 | 0.02963 | 0.02776 | 0.02814 | 0.02762 | 0.02753 | 0.02732 | 0.02564 | 0.02602 |
> | Omniglot  |   S  | 0.03038 | 0.03627 | 0.03078 | 0.03068 | 0.02714 | 0.02967 | 0.02889 | 0.02668 | 0.02631 |
> |              |   L  | 0.02704 | 0.03192 | 0.02728 | 0.02696 | 0.02567 | 0.02648 | 0.02644 | 0.02578 | 0.02539 |
> |   SVHN    |   S  | 0.00320 | 0.00420 | 0.00320 | 0.00369 | 0.00273 | 0.00317 | 0.00307 | 0.00202 | 0.00192 |
> |              |   L  | 0.00174 | 0.00204 | 0.00190 | 0.00177 | 0.00178 | 0.00173 | 0.00175 | 0.00148 | 0.00147 |
> |  CIFAR10  |   S  | 0.01466 | 0.01620 | 0.01431 | 0.01427 | 0.01208 | 0.01452 | 0.01504 | 0.00768 | 0.00691 |
> |              |   L  | 0.00960 | 0.01123 | 0.00863 | 0.00900 | 0.00755 | 0.00832 | 0.00898 | 0.00629 | 0.00587  |
> | CIFAR100 |   S  | 0.01465 | 0.01713 | 0.01463 | 0.01484 | 0.01369 | 0.01391 |  0.01477 | 0.00765 | 0.00717 |
> |              |   L  | 0.01015 | 0.01064 | 0.00951 | 0.00862 | 0.00842 | 0.00910 |  0.00912 | 0.00678 | 0.00635 |
> |  CELEBA   |   S  | 0.00780 | 0.00937 | 0.00830 | 0.00782 | -          |            | 0.00861 | 0.00608  | 0.00747 |
> |              |   L  | 0.00613 | 0.00646 | 0.00630 | 0.00590 | -          |            | 0.00665 | 0.00563 | 0.00565 |
>
>
> We will continue to update the table (and our manuscript) as we obtain further experimental results, but the preliminary results that we've obtained so far indicate that both NRAE-L and NRAE-Q outperform GRAE.
>
> **Q2.** The authors also do not compare to SPAE in Figure 9. This
> should be corrected as in Figure 8, SPAE was the only other method that
> correctly learned the manifold.
>
> **A2.** We agree that it is better to include the results of SPAE in Figure 9 for completeness. We will add this in the updated manuscript.
>
> **Q3.** The authors also claim that "it is in fact the decoder,
> and not the encoder, that has a greater impact on learning the correct
> manifold". The authors do not sufficiently justify this. They show
> that their approach seems to work well, but that is not sufficient
> justification for their claim. For one thing, regularizing the bottleneck
> has a direct effect on the decoder as well as it changes the inputs to it.
> Thus it is incorrect to claim that any method that only regularizes the
> bottleneck is ignoring the decoder. The authors should do more experiments
> demonstrating very clearly that focusing on the output of the decoder is
> superior to other methods that perform regularization on other parts of
> the autoencoder.
>
> **A3.**
> We believe the following clarification on the terminology and expressions will be useful in the subsequent discussion.
>
>
> * By "regularization" we mean the inclusion of a regularization term in the loss function used for autoencoder training.
> *  The expression "regularizing the encoder only" means that the regularization term only involves the encoder mapping.
> * The expression "regularizing the decoder only" means that the regularization term only involves the decoder mapping.
> * "Bottleneck" refers to the latent space. "Regularizing the bottleneck" we then understand to mean that the regularization term involves the encoder only (that is, letting $z=g(x)$ be the encoder mapping, then only $g$ is included).
>
> With the above fixing of terminology and expressions, a more precise
> statement of our claim would be that regularization terms that involve both the encoder and decoder mappings are important than those that only involve the encoder mapping.
>
> To be clear, we are **not** making the claim that any method that only regularizes the latent space distribution is "ignoring" the decoder. A more accurate statement of our assertion would be that encoder-only regularization methods alone are insufficient to mitigate overfitting; we take into account both the encoder and decoder, and through our analysis arrive at the conclusion that "matching the local geometry of the decoded manifold to that of the original data distribution" is very important to resolve the two main issues discussed in the paper.
>
> In retrospect, it seems that some of our statements made in
> the original manuscript were insufficiently precise, most notably our assertion
> that (line 48) "...it is in fact the decoder, and not the encoder,
> that has a greater impact on learning the correct manifold".
>
> Further changes planned for the updated manuscript include:
>
> * (line 48) it is in fact the decoder, and not the encoder,
> that has a greater impact on learning the correct manifold $\mapsto$
> matching the local geometry of the decoded manifold to that of the original data distribution significantly improves the reconstruction quality (i.e., learning the correct manifold)
> * (line 6) The focus of these methods is **exclusively** on the encoder
> $\mapsto$ The focus of these methods is on the encoder
> * (line 8, 170) neglecting the decoder can lead to noise sensitivity and overfitting -> **using the regularization term that does not depend on the decoder** can lead to noise sensitivity and overfitting
>
> Lastly, we want to note that as shown in Table 3, combining NRAE with other existing encoder regularization methods leads to better performance than using NRAE alone.

---

> > ### Comment · Reviewer_JTcy · 2021-08-23
> > **Concerns largely addressed**
> >
> > I appreciate the authors' response. With the proposed changes and experimental additions, I am willing to change my score to accept.

---

### Official Review · Reviewer_mRsa · 2021-07-20

**Rating:** 7
**Confidence:** 3

**Summary:**

This paper proposes a new graph-based autoencoder dubbed ‘Neighborhood Reconstructing Autoencoder’ (NRAE) for learning smoother manifolds. The goal is to (1) learn an AE that is robust to data noise (2) learn a manifold with the correct geometry. The authors claim the focus of regularization to induce a smooth representation should, unlike the previous work, be on the decoder and not the encoder. The proposed model minimizes an objective that encourages neighbouring xs (based on a predefined graph that is constructed using some kernel K(x,x’)) to also be close by in latent space by having a small ‘neighbourhood reconstruction loss’. The authors finally perform an extensive set of experiments on a variety of datasets which demonstrates that the proposed model is able to learn a more smooth manifold compared to other AE baselines, as well as having a better generalization error.

**Limitations And Societal Impact:**

The authors did a find job discussing the limitations and potential negative societal impact of their work.

**Main Review:**

[General Comment]

I enjoyed reading the paper. I think the idea behind the proposed approach certainly makes sense and can be useful in many domains. The extensive number of experiments in various scenarios makes the paper even stronger. While I think there are few things that need clarifying, I think this paper makes a good contribution to NeurIPS and I’m happy to recommend acceptance.

- The authors could correct me if I’m wrong but I think the implicit assumption in this paper is that the ‘correct’ manifold is also in fact a 'smooth' one (therefore proposing to minimize the neighborhood reconstruction loss). In many real-world settings however, the ground-truth manifold is not smooth at all. Could the authors comment on this?
- Similarly, distinguishing noise from features of interest is also an ill-posed problem as far as I know (in a fully unsupervised manner). That being said, the kernel we use here can act as inductive bias as to what we consider to be ‘neighbour’ in data space which I find very helpful. I think, if there is enough space, the authors should provide more discussion on this.
- While I think I understand the claim of focusing on regularizing the decoder instead of encoder, I’m not sure if there is enough explanation in the current draft about ‘why’ this is the case. The authors explain in paragraph 3 in the intro that decoder is also susceptible to overfitting which makes sense, but this doesn't explain why only regularizing the decoder and not encoder works significantly better. The decoder overall has a more difficult task as it has to map low-dimensional to high dimensional data, and in the cases where the training set is large, wouldn't it be very hard for the decoder to overfit?
- Could the authors discuss the connection of the proposed method to [1]?
- I understand kNN is one of the simplest & common methods to construct graphs but euclidean distance in high-dimensional domains such as vision is typically not a good way to quantify similarity, especially if the data is sparse. To put it another way, there is no strict reason for why two points that are close in x-space (w.r.t L2 distance) should also be close in latent space. I understand that addressing this issue might be beyond the scope of this paper given that this is a fundamental problem, but I would still appreciate it if the authors could comment on it.
- The experiment Section in particular was very good I think. The experiments done here are very extensive and I think are designed very well, perfectly making the case for learning smoother manifolds and their benefit in terms of reconstruction error.

[Clarity]

This paper is very well written. Most of the figures were very well designed and informative.

[Novelty & Prior work]

I am not very familiar with the field, but the discussion Related work section seems sufficient to me. The few papers that I think should be cited but not cited here are [1-3]


[Minor Comments/Questions]

- In Figure 4, what happens if you simply use weight regularization to smooth the manifold of a vanilla autoencoder; AE_LossL + λ||w||?

- Figures 4, 6, & 7, and Table 2 are all very good!

- I’m curious about the Experiment in Figure 9. Sorry if I missed it but could the authors elaborate on ‘training with a circular latent space’? How was the latent space designed here exactly?



[References]

[1] Yeh, Chin-Chia Michael, et al. "Representation Learning by Reconstructing Neighborhoods." arXiv preprint arXiv:1811.01557 (2018).

[2] Wang, Wei, et al. "Generalized autoencoder: A neural network framework for dimensionality reduction." Proceedings of the IEEE conference on computer vision and pattern recognition workshops. 2014.

[3] Liang, Kongming, et al. "Representation learning with smooth autoencoder." Asian conference on computer vision. Springer, Cham, 2014.


**Time Spent Reviewing:**

4

---

> ### Author Response · Authors · 2021-08-10
> **Response for Reviewer mRsa**
>
> Dear Reviewer mRsa,
>
> Thank you for your constructive feedback. We are glad that you found our paper enjoyable.
>
> **Q1.** The authors could correct me if I’m wrong but I think the implicit assumption in this paper is that the ‘correct’ manifold is also in fact a 'smooth' one (therefore proposing to minimize the neighborhood reconstruction loss). In many real-world settings however, the ground-truth manifold is not smooth at all. Could the authors comment on this?
>
> **A1.** You are correct, the implicit assumption is that the correct manifold
> is indeed smooth. As you also correctly note, this assumption may not always hold
> in real-world settings: the manifold may have boundaries, self-intersections,
> or even be a union of multiple manifolds with different dimensions and topologies.
> Trying to address all of these myriad cases requires many assumptions and
> engineering choices, and we feel is too much of a diversion from the main
> message of our paper. Certainly, these topics are important ones that we hope
> to address in future follow-up work.
>
> **Q2.** Similarly, distinguishing noise from features of interest is also an ill-posed problem as far as I know (in a fully unsupervised manner). That being said, the kernel we use here can act as inductive bias as to what we consider to be ‘neighbour’ in data space which I find very helpful. I think, if there is enough space, the authors should provide more discussion on this.
>
> **A2.** Yes, we agree. I'm not sure if there's enough space for a lengthy discussion on this topic, but we certainly intend to point out
> in the revised manuscript that in our approach, both the choice of kernel and the graph construction can be viewed as inductive biases.
>
> **Q3.** While I think I understand the claim of focusing on regularizing the decoder instead of encoder, I’m not sure if there is enough explanation in the current draft about ‘why’ this is the case. The authors explain in paragraph 3 in the intro that decoder is also susceptible to overfitting which makes sense, but this doesn't explain why only regularizing the decoder and not encoder works significantly better. The decoder overall has a more difficult task as it has to map low-dimensional to high dimensional data, and in the cases where the training set is large, wouldn't it be very hard for the decoder to overfit?
>
> **A3.** First, we should clarify that our NRAE regularizes both the encoder and decoder,
> and not only the decoder as we may have unintentionally implied in our original
> manuscript. It appears that the statement (line 48) that ''...it is
> in fact the decoder, and not the encoder, that has a greater impact on
> learning the correct manifold" may have been the source of this unintended
> implication.  In the updated manuscript we've tried to make this statement
> more precise as follows: "matching the local geometry of the decoded manifold to that of the original data distribution significantly improves the reconstruction quality (i.e., learning the correct manifold)"
>
> **Q4.** In Figure 4, what happens if you simply use weight regularization
> to smooth the manifold of a vanilla autoencoder; AE-Loss $L + \lambda\|w\|$?
>
> **A4.** We have performed experiments with the L1 regularizer using
> the example in Figure 4. Interestingly enough, we have discovered that
> the L1 regularizer manages to effectively smooth out the noise to some
> degree. Our qualitative judgment is that the smoothing performance of L1 is
> comparable to SPAE and somewhat better than AE, VAE, or WAE. However,
> its performance is clearly less satisfying than DAE or NRAE.
>
> **Q5.** I’m curious about the Experiment in Figure 9. Sorry if I missed
> it but could the authors elaborate on ‘training with a circular latent space’?
> How was the latent space designed here exactly?
>
> **A5.** We'll try to clarify this point in the updated manuscript,
> but we simply add one layer to the end of the encoder corresponding to
> $z\mapsto \frac{z}{\|z\|}$.
>
> **Q6.** Could the authors discuss the connection of the proposed method to [1]? The few papers that I think should be cited but not cited here are [1-3]
>
> **A6.** Thank you very much for introducing the related works. We will discuss how they are related to our work in the updated version of the manuscript. In particular, [1] proposes "the neighbor-encoder" which reconstructs the input data's neighbors, where $k$ decoders are trained for $k$ neighborhood points. In my opinion so far, it is difficult to find a clear connection point, except for the common point of using a neighborhood graph.
>
> **Q7** I understand kNN is one of the simplest & common methods to construct graphs but euclidean distance in high-dimensional domains such as vision is typically not a good way to quantify similarity, especially if the data is sparse. To put it another way, there is no strict reason for why two points that are close in x-space (w.r.t L2 distance) should also be close in latent space. I understand that addressing this issue might be beyond the scope of this paper given that this is a fundamental problem, but I would still appreciate it if the authors could comment on it.
>
> **A7** We agree with your opinion. Besides L2 distance, different distance metrics that better reflect the data similarity can certainly benefit the graph construction step. As an example, domain knowledge can be used to define the similarity between data. We believe that this is related to the discussion of inductive bias in **A2**. We will try to point out in the revised manuscript that in our approach different distance metrics can be used for the graph construction.
>
> We thank you again for your efforts in reviewing our paper!
>
> Best regards,
> Authors.

---

> > ### Comment · Reviewer_mRsa · 2021-08-18
> > **Thank you for your detailed response. Still happy to recommend acceptance**
> >
> > Thank you for your comments,
> >
> > I have read the response and other reviews.
> >
> > Thank you for the clarification regarding the Q3 and Q5. I still encourage the authors to add some of the discussion presented in A2 and A7 in the main paper as I think it will make the paper stronger. I'm still happy to recommend acceptance.
> >
> > Best

---

> > > ### Author Response · Authors · 2021-08-23
> > > **Response for Reviewer mRsa**
> > >
> > > Thank you! We will try to add those discussion points in the revised manuscript!

---

### Decision · Program_Chairs · 2021-09-27

**Decision:**

Accept (Poster)

**Comment:**

This work introduces NRAE -- Neighborhood Reconstructing Autoencoder. The high level goal is regularization to avoid overfitting to noise and learning a smooth manifold with the 'correct' geometry. The authors introduce an objective that encourages a small neighbourhood reconstruction loss’ based on a kernel. The paper is easy to follow, clearly written and includes an extensive set of experiments on a variety of datasets.


During the rebuttal, the authors were able to successfully address the concerns raised by the reviewers, in particular, comparisons with GRAE, SPAE and nuanced discussion of the claims about decoder regularization. There is a consensus among the reviewers that the paper deserves acceptance.